# Source-Free Controlled Adaptation of Teachers for Continual Test-Time Adaptation

## Abstract

In many real-world scenarios, encountering continual shifts in domain during inference is very common. Consequently, continual test-time adaptation (CTTA) techniques leveraging a teacher-student framework have gained prominence, allowing models to adapt continuously even after deployment. In such a framework, a weight-averaged mean teacher is used to produce pseudo-labels from test data for self-training. The mean teacher gets updated as an exponential moving average of the student parameters using a high value of momentum that is kept fixed even if different distributions of test data are encountered. To combat the resulting drift of the model, we propose a novel controlled teacher adaptation methodology that dynamically sets a proper momentum value depending on the quality of the incoming data. Additionally, we estimate class prototypes from the source pretrained model to help align the target data as they come in. Importantly, our method does not require access to source data or its statistics at any stage of the pipeline, making it truly source-free. We perform extensive experiments on benchmark datasets to demonstrate that our approach outperforms different state-of-the-art adaptation frameworks, many of which require access to source data.

## 1 Introduction

Deep Neural Networks have demonstrated remarkable representation and generalization capabilities on various scene understanding tasks. While the promise is certainly there, the real-life performance of many of these methods falls significantly when faced with distributional shifts in applications. This is because data in the domain where the models are deployed (*target domain*) is not distributed identically to the training data in the domain where they are trained (*source domain*). To address this gap, it is often necessary to adapt a source pre-trained network to the target domain without any supervision from the target domain (known as unsupervised domain adaptation, UDA) (Araslanov & Roth, 2021; Ganin et al., 2016; Hoffman et al., 2018; Long et al., 2015; Mei et al., 2020; Sahoo et al., 2021; Tzeng et al., 2017). Current UDA approaches assume that labeled source data and unlabeled target data are available during adaptation. However, both these assumptions can be unrealistic in many scenarios. Although pre-trained models are easily available nowadays, the source data used for training these are often not available due to privacy, storage or financial constraints. Moreover, for an already deployed model, it may be imperative not to wait long to collect data from the new domain as inference must continue. To address this challenge, Test-Time Adaptation (TTA) (Niu et al., 2022; Shin et al., 2022; Wang et al., 2021) has emerged as a promising approach.

Existing TTA approaches rely on a restrictive assumption that the target domain is isolated and stationary. However, in real-world scenarios, the target domain can continually evolve. For example, a model trained with data from clear weather conditions, may need to work on-the-fly in diverse weather conditions such as snow, rain, fog or haze. To address the continual drift in data distribution in absence of source data, researchers have started to explore continual test time adaptation (CTTA) methods (Chakrabarty et al., 2023; Döbler et al., 2023; Niloy et al., 2024; Wang et al., 2022; 2024). Typically, CTTA approaches adapt the model by updating its parameters during the test phase via self-training. This is done by employing a teacher-student setup, where the student model acts as the primary model, trained using pseudo-labels that are generated by the teacher model. In the continually changing environment, the model may gradually shift

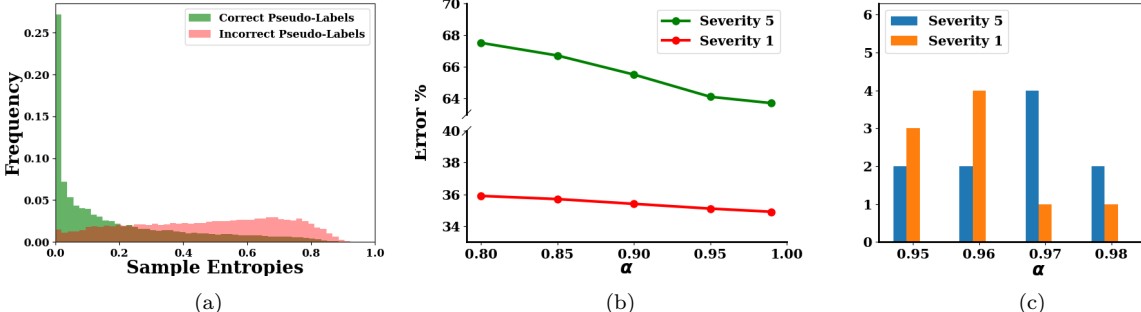

Figure 1: To justify our design choices, we conducted three experiments on the ImageNet-C dataset. (a) The frequency distribution of entropy values of $75,000$ images spanning over 15 types of corruptions. Green distribution is for the samples with correct pseudo-labels while the red distribution is for the incorrect ones. Samples correctly predicted are likely to have lower entropy predictions. (b) Average error across all 15 noise types with different levels of severity (corruption severity levels 1 and 5) with the RMT (Döbler et al., 2023) model. $x$ axis lists different fixed momentum ($\alpha$) values with which the RMT teacher is updated. Performance varies differently with the choice of $\alpha$ depending on the distribution shift. (c) The number of noise types achieving minimum error rates vs. $\alpha$ demonstrates that performance is optimal at different $\alpha$ values for different noise types, highlighting the need for a method to compute $\alpha$ dynamically. (Best viewed in color.)

and thus the pseudo-labels can become progressively noisier. Such mis-calibrated samples, when used in further adaptation, can lead to error accumulation.

Motivated by the success of weight-averaged models in self-supervised learning (Polyak & Juditsky, 1992; Tarvainen & Valpola, 2017), recent CTTA approaches have leveraged a weight-averaged teacher (Döbler et al., 2023; Wang et al., 2022). The student model is continuously updated using pseudo-labels generated by the teacher. The teacher is updated using an exponential moving average (EMA), where a *momentum value $\alpha$* controls the influence of the current batch on the running average. A low value of $\alpha$ incurs a drastic change to the teacher, while a high value more or less maintains the status quo. Ideally, if data from the current domain is drastically different, then the generated pseudo-labels are noisy and unreliable. The model, naturally, gets confused, and this is manifested by the increased entropy of the prediction by the teacher model. As shown in Fig. 1a, samples that give incorrect pseudo-labels tend to produce higher entropy compared to those with correct pseudo-labels. Thus, in contrast to previous works (Döbler et al., 2023; Wang et al., 2022) which use a fixed momentum, we propose to adaptively choose higher or lower momentum values depending on the prediction entropy of a batch. By dynamically adjusting the momentum, the teacher model strikes a balance between adapting to distribution shifts and maintaining stability, leading to improved performance.

An important drawback of many recent CTTA approaches is that they often remain dependent on source data. For example, RMT (Döbler et al., 2023) and SANTA (Chakrabarty et al., 2023) utilize source data to establish class-wise source prototypes for warm-starting the adaptation process. While this technique helps in achieving meaningful clustering and good class separation in unseen domains, it requires access to the source data and thus such approaches can not be regarded as truly source-free. To effectively tackle this, we employ an alternate approach to estimate the source class prototypes by utilizing the pre-trained model itself. Specifically, we treat the weights learned by the classifier in the last layer of the source pre-trained models as the class prototypes. As the dot product of the features and last layer weights to a particular output neuron determines the score of the corresponding class, the weights are aligned with the features of the class. Hence, we use the weight vectors from the classifier for each output neuron as the source class prototypes. By leveraging the source pre-trained model only, our approach eliminates the need for source data at any stage of the framework. After warm-starting, the class prototypes are updated with confident target domain samples to incorporate valuable domain-specific information with continually changing domains.

Our proposed approach `DMSE` (Dynamic Momentum and Source Estimation) dynamically updates the model and harnesses the pre-trained model towards source-free CTTA. Extensive experiments on four benchmark datasets demonstrate the superiority of our method over the state-of-the-art, including ones requiring access

to source data. We perform extensive ablations to depict the importance of each component of the framework. Our contributions include:

- We propose a dynamic momentum update based on the average prediction entropy enabling the teacher to adapt to distribution shifts, leading to better CTTA performance.
- Unlike existing approaches, we leverage the classifier itself to estimate source prototypes without requiring access to the source domain data at all during adaptation.
- Extensive experiments and ablations over multiple benchmark datasets, showing consistent benefits of `DMSE` (implementation to be made public) over SOTA.

## 2 Related Works

**Unsupervised Domain Adaptation**: Unsupervised Domain Adaptation (UDA) adapts a source pre-trained model to a target domain when data from the source model is available, and data from the target domain is also available but without labels. Traditionally, UDA approaches align source and target data by minimizing domain discrepancy (Chen et al., 2020a; Shen et al., 2018; Sun & Saenko, 2016) or maximizing domain confusion (Liu et al., 2021a; Long et al., 2018; Tzeng et al., 2017). Recently, self-supervised approaches *e.g.*, contrastive learning (Li et al., 2020a; Prabhu et al., 2021; Sahoo et al., 2021), solving pretext tasks (Carlucci et al., 2019; Mei et al., 2020) and pseudo-labels (Chen et al., 2019; Sahoo et al., 2023; Xie et al., 2018) have been applied in aligning domains. These are especially popular in adapting domains source-free, where source data is inaccessible (Ahmed et al., 2021; Ding et al., 2022; Kumar et al., 2023; Liang et al., 2020; Xia et al., 2021). Some existing works adapt without source data relying on generative modeling (Kurmi et al., 2021; Li et al., 2020b).

**Test-Time Adaptation**: Traditionally, UDA methods are dependent on huge amount of target domain data regardless of whether source data is utilized. Once deployed, such models are incapable of training under changing scenarios before new target domain data can be collected. Test Time Adaptation (TTA) is a variant that leverages test samples encountered in the target domain after deployment to adapt the source pre-trained model. A popular direction is to adjust some of the model parameters by minimizing unsupervised loss functions on the unlabeled test samples. TENT (Wang et al., 2021) updates the batch-norm statistics of the pre-trained model by minimizing the entropy of the predictions. Authors in (Iwasawa & Matsuo, 2021) train only the final classification layer with pseudo-prototypes from the test data. Some approaches (Liu et al., 2021b; Sun et al., 2020) introduce additional self-supervised tasks during source training. During testing, this additional module is adapted on test data from the target domain. SHOT (Liang et al., 2020), uses source data to train a specialized module using diversity regularizer with label smoothing in addition to entropy minimization. Naturally, the reliance of this paradigm on additional model modifications in both training and inference phases, makes it impractical and non-scalable in real-world scenarios.

**Continual Test-Time Adaptation (CTTA)**: While adapting to a single target domain presents a challenge in itself, a more realistic scenario presents the need for continual adaptation to a series of domain shifts. There have been attempts to apply TTA approaches on the CTTA setting as well. However, vanilla TTA methods (Mirza et al., 2022; Wang et al., 2021) when applied in this setting, suffer from *error accumulation* by continually drifting away from source knowledge. Recent works try to address this challenge by proposing targeted techniques to overcome the error accumulation. CoTTA (Wang et al., 2022) introduced a self-training technique using augmentation averaged predictions between a moving average teacher and student model. RMT (Döbler et al., 2023) makes use of a symmetric cross-entropy in a teacher-student framework, coupled with a contrastive loss to bring the test feature space closer to the source feature space. SANTA (Chakrabarty et al., 2023) removes the requirement of maintaining a teacher model and uses source anchoring for self-training. EATA (Niu et al., 2022) introduces weight regularization to keep the adapted weights close to the source pretrained model. Authors in (Niloy et al., 2024) use batch-norm statistics of the incoming batches to detect domain change and modulate model resets. Most of these works require the source data at some stage or do not follow the fully online setting. We focus on the fully test-time setting where, instead of using source data we make use of the pretrained classifier to get the source prototypes and dynamically adjust the momentum parameter of the teacher to gracefully handle model drift due to error accumulation.

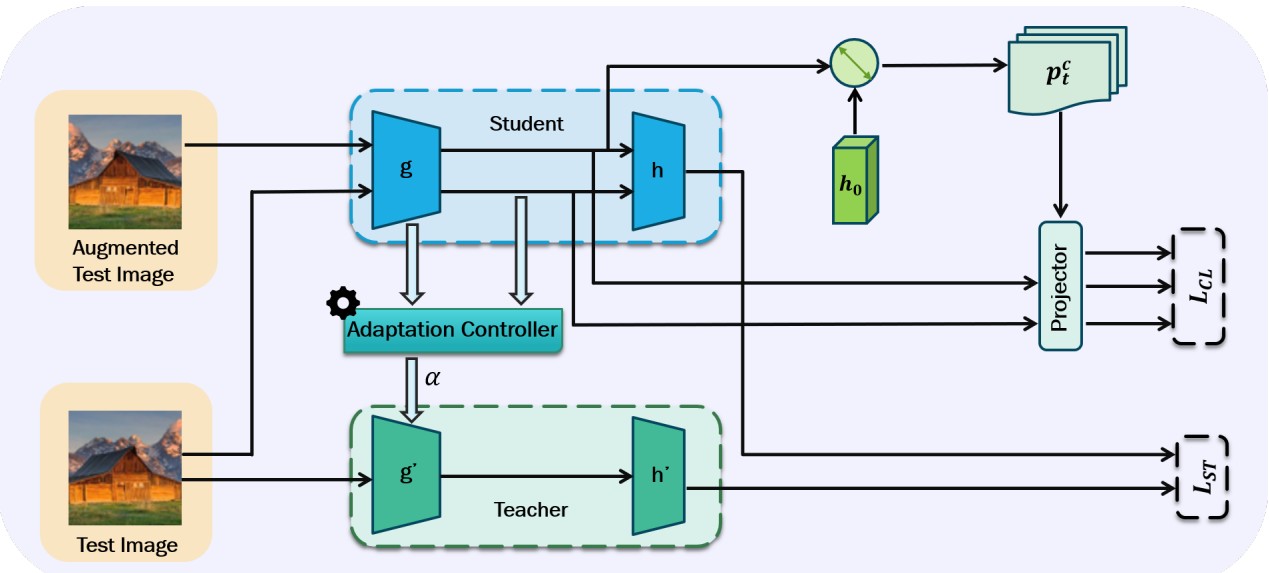

Figure 2: **The proposed DMSE architecture:** The student model is trained with pseudo labels from the mean teacher. The teacher is updated using EMA from the student with a dynamically determined $\alpha$ based on the student's prediction entropy. If the entropy falls below a threshold, the teacher model resets to the source model. Additionally, class-wise prototypes are dynamically updated using confidently pseudo-labelled test data. For inference, a summation of outputs of both the student model and the teacher model is considered.

## 3 Methodology

In the Continual Test-Time Adaptation (CTTA) setting, given a source pre-trained model $f_{\theta_0}$, we have to continually adapt this pre-trained source model to a sequence of varying target domains $\{D_k\}_{k=1}^K$, where $K$ is the total number of target domains. The test samples arrive in an online fashion and are encountered by the learner only once. At each time-step $t$, the learner encounters test samples $x_{t,k}$ from domain $D_k$. The learner must make predictions $f_{\theta_t}(x_{t,k})$ on the encountered test samples, $x_{t,k}$, and adapt itself ($f_{\theta_t} \to f_{\theta_{t+1}}$) for the test samples yet to come, in the future timesteps. Furthermore, in our fully test-time adaptation setting, source data is not available for use at any point. This decision stems from concerns about data privacy and unavailability in real-life scenarios.

In this work, we propose a source-free continual test-time adaptation approach that addresses the challenges of adapting to distribution shifts while maintaining model performance. Our approach employs a controlled teacher adaptation mechanism, enabling the teacher model to adapt to changing distributions while preserving its robustness. Additionally, we estimate class-wise prototypes from the source pre-trained model to form disentangled clusters for unseen domains, further enhancing the model's ability to generalize to new environments. The overall scheme of the proposed approach is shown in Fig. 2. In the subsequent subsections, we delve into the details of the controlled teacher adaptation and class-wise prototype estimation.

### 3.1 Controlled Teacher Adaptation

Self-training a network by using its own predictions as pseudo labels has proven to be very effective in semi-supervised learning and unsupervised domain adaptation (Manohar et al., 2018; Sahoo et al., 2023; Sohn et al., 2020). Vanilla self-training methods using pseudo-labels (Lee, 2013; Wang et al., 2021) thrive when the pseudo-labels are reliable as a result of more or less unchanging data distribution. However, in CTTA with continually changing target domains, the distribution shift results in noisy pseudo-labels and self-training with them leads to error accumulation. The mean-teacher framework (Tarvainen & Valpola, 2017) has been employed by existing CTTA approaches (Döbler et al., 2023; Wang et al., 2022; 2024) to produce pseudo-labels and mitigate accumulation of errors to some extent. A mean teacher in a student-teacher

framework has no gradient flowing through it and shares the same architecture as the student model. Its parameters get updated using exponential moving average (ema) over the current teacher parameters and the updated student parameters. Mathematically,

$$\theta'_{t+1} = \alpha \cdot \theta'_t + (1 - \alpha) \cdot \theta_{t+1} \tag{1}$$

where $\theta$ and $\theta'$ are the student and teacher parameters respectively with the subscripts denoting the timesteps. $\alpha \in [0, 1]$ is the momentum value that controls the influence of the student model on the weight updates in the current teacher model. A low value of $\alpha$ allows the teacher model to adapt more readily to the changing data distribution, but it also risks adapting too much to a student model which can be detrimental especially if incorrect pseudo-labels are prevalent. Existing teacher-student frameworks tend to rely on a fixed and high value of $\alpha$. However, using a high $\alpha$ not only limits the adaptability of the teacher model to an evolving data distribution, but also a fixed momentum value may lead to sub-optimal performance, as we show below.

**Demonstrating the problem with a fixed $\alpha$**: To this end, we conduct an experiment using an ImageNet pretrained ResNet-50 model. We took a SOTA teacher-student framework RMT (Döbler et al., 2023) and presented corrupted test images from the ImageNet-C dataset (Hendrycks & Dietterich, 2019) after applying 15 different types of corruptions. We experimented with the highest and lowest corruption severity levels (5 and 1 respectively) for this purpose with different fixed values of $\alpha$ ranging from 0.8 to 0.999. Fig. 1b shows how the performance (average error across 15 noise types) varies with the momentum ($\alpha$) values when the teacher model is updated with these fixed $\alpha$'s. A high noise severity implies less reliable pseudo-labels and thus high momentum values help. However, for less severe noise, the data distribution does not change much and the pseudo-labels are more reliable. As the change in data distribution is low, the student sees very similar data to what the teacher has seen till now and thus there is very little difference between the two models. As a result, the optimal performance is indifferent to whether the new teacher is influenced more by the current teacher (high $\alpha$) or the current student (low $\alpha$) as shown by the nearly constant performance across the whole range of $\alpha$ (ref. Fig. 1b). This experiment shows that the optimal momentum value can be different depending on the type of data the model encounters. While we do not deny that a higher momentum, on average, gives a lower error over different sets of corruptions, we emphasize that it isn't necessary that one fixed momentum value would give best performance for every noise over a sequence of corruptions. This is further shown in Fig. 1c which shows that different $\alpha$ values are optimal for different noise types depending on the severity of the noises. Detailed results for individual noise types are provided in the appendix.

**Addressing the problem:** To tackle this, we propose a controlled momentum variation approach where the extent of knowledge transfer between the student and the teacher models would be adjusted on the basis of the quality of incoming test batches. The distribution shift and the subsequent reliability of the generated pseudo-labels are manifested by the entropy of the prediction by the teacher. When the underlying distribution of the data changes significantly, it causes a noticeable increase in the prediction entropies. So, we propose to adjust the $\alpha$ value depending on the entropy, where a test batch with lower entropy is assigned a lower $\alpha$ (*i.e.*, more knowledge transfer from the student model) and vice-versa. Specifically, for the average prediction entropy $e$ of a batch by the student model, we calculate $\alpha$ as follows:

$$\alpha = \min(\alpha_{min} + e \cdot \beta, 1) \tag{2}$$

where $\alpha_{min}$ is a hyper-parameter that denotes the minimum value of $\alpha$ and $\beta$ is the scaling factor. Additionally, to maintain stability and prevent potential collapse in the teacher model, we incorporate a resetting strategy inspired by (Niu et al., 2023). This strategy involves resetting the parameters of the teacher model to the original pre-trained weights. The resetting is triggered when the prediction entropy of the student model drops below a specified entropy threshold $e_{min}$, since this serves as an indicator of overconfidence and potential overfitting to recent data.

## 3.2 Class-wise Prototype Estimation

Recent works have resorted to using source data either partially to counter the effect of domain shift during test time (Niu et al., 2022) or fully to fetch source class-wise prototypes to warm up the model before

adaptation (Chakrabarty et al., 2023; Döbler et al., 2023). While class-wise prototypes help in target alignment, requiring access to the source data at any stage of the pipeline is a privilege and narrows down the applicability of such approaches. Hence, we rely on the source pre-trained classifier to estimate the source prototypes not requiring access to the source data.

We denote the source pre-trained model as $f_{\theta_0}$, where the subscript 0 indicates the initial time step. For notational convenience, we drop the subscript and refer to it simply as $f_\theta$. The source pre-trained model is a composition of a CNN feature extractor ($g$) and a linear classifier ($h$), $i.e.$, $f_\theta = h(g)$. The input $x$ goes through the feature extractor $g$ to generate features $\mathbf{g}_x \in \mathbb{R}^d$, which, in turn, goes through the classifier to obtain class-wise logits. Specifically, for $C$ classes, considering $\mathbf{W}_h \in \mathbb{R}^{C \times d}$ as the weight matrix of the classifier, the prediction $\hat{\mathbf{y}} \in \mathbb{R}^C$ is given by, $\hat{\mathbf{y}} = \mathbf{W}_h \mathbf{g}_x$. Each element in $\hat{\mathbf{y}}$ is a result of the dot product between a row (vector) of the weight matrix $\mathbf{W}_h$ and the feature vector $\mathbf{g}_x$. Ideally, features from an image belonging to a class $c$ will have the highest dot product value with the $c^{th}$ row of $\mathbf{W}_h$. This suggests that normalized features from images belonging to the $c^{th}$ class tend to cluster around this vector, making it a good candidate for a prototype for that class, in absence of source data. In our work, these $C$ row vectors from $\mathbf{W}_h$ act as the initial class prototypes $p_0^c$ where the subscript 0 corresponds to the initial time-step.

While these class prototypes help in the initial alignment of domains, as the shift in data distribution is continual in CTTA, the prototypes need to be updated with newly arriving data, otherwise, target features would drift away from the class prototypes. Unlike existing approaches (Chakrabarty et al., 2023; Döbler et al., 2023), we propose to dynamically update these class-wise prototypes to improve alignment with target features, particularly in cases of significant domain variations. Let the $i^{th}$ input sample at timestep $t$ be denoted as $x_i$. Note that the current CNN feature extractor is a result of update from the previous timestep $t-1$ and is denoted by $g_{t-1}(\cdot)$. So, the feature generated at the current timestep is $g_{t-1}(x_i)$. We compute the cosine distance $dist(g_{t-1}(x_i), p_{t'}^c)$ between the test samples $(x_i, \forall i)$ and the class prototypes $(p_{t'}^c, \forall c)$, where $t'$ is a previous timestep compared to $t$. The distance is computed as $0.5(1 - cos(g_{t-1}(x_i), p_{t'}^c))$ where $cos(x, y)$ denotes cosine similarity. The factor 0.5 scales the cosine distance to lie within the $[0, 1]$ range. After getting these distances, we find the closest class prototype to the sample as, $\hat{c}_i = \underset{\forall c}{\operatorname{argmin}}\, dist(g_{t-1}(x_i), p_{t'}^c)$. The sample $x_i$ is assigned a pseudo-label $\hat{c}_i$. The average feature of all the samples having the same pseudo-label provides the updated prototype of that class at the current timestep. However, instead of blindly believing all samples, we consider only those samples that are close enough to their assigned class in the feature space. Mathematically for $t' < t$,

$$p_t^c = \frac{\sum\limits_{i \text{ with } \hat{c}_i = c} g_{t-1}(x_i) \mathbb{1}(dist(g_{t-1}(x_i), p_{t'}^c) < \gamma)}{\sum\limits_{i \text{ with } \hat{c}_i = c} \mathbb{1}(dist(g_{t-1}(x_i), p_{t'}^c) < \gamma)} \tag{3}$$

$\gamma$ is the threshold to filter out the samples as described above. Specifically, we experimented with two separate settings of $p_{t'}^c$. We used the initial class prototypes $p_0^c$ and the immediately previous class prototypes $p_{t-1}^c$ in the right hand side of the equation above to get the updated class prototypes $p_t^c$. Our ablation study (ref. Table 9) shows that using the initial class prototypes ($i.e.$, using $t' = 0$) helps more.

### 3.3 Final Objective

In line with (Döbler et al., 2023; Chakrabarty et al., 2023), we use two losses – a) symmetric cross-entropy loss (Wang et al., 2019) and b) contrastive loss (Khosla et al., 2020). The symmetric cross-entropy loss between two distributions $p$ and $q$ with $C$ elements is,

$$\mathcal{L}_{\text{SCE}}(q, p) = -\sum_{c=1}^{C} q_c \log p_c - \sum_{c=1}^{C} p_c \log q_c, \tag{4}$$

For an input $x$, we compute $\mathcal{L}_{\text{SCE}}$ by comparing the softmax predictions of the teacher model ($f_{\theta'}(x)$) and the student model ($f_\theta(x)$). To enhance prediction stability against slight changes, we compute the symmetric cross-entropy loss between $f_{\theta'}(x)$ and predictions made on a randomly augmented version $\tilde{x}$ by the student

| Time | $t \xrightarrow{\hspace{4cm}}$ | | | |
|------|---------|----------|--------|------|
| Method | clipart | painting | sketch | Mean |
| CoTTA | 45.2 | 35.7 | 49.2 | 43.4 |
| RDumb | 43.2 | 35.0 | 46.8 | 41.7 |
| SANTA | 38.8 | 34.1 | 43.8 | 38.7 |
| RMT | 37.8 | 32.4 | 42.6 | 37.6 |
| DMSE | 38.3 | 32.1 | 41.9 | **37.4** |

Table 1: **Classification error rate (%) DomainNet-126** (with the real domain as source domain). Note that both RMT and SANTA require access to the source data at the start of adaptation.

model, represented as $f_\theta(\tilde{x})$. This process yields a self-training loss as follows:

$$\mathcal{L}_{ST} = \frac{1}{2}(\mathcal{L}_{SCE}(f_\theta(x), f_{\theta'}(x)) + \mathcal{L}_{SCE}(f_\theta(\tilde{x}), f_{\theta'}(x))) \tag{5}$$

The contrastive loss brings a test example closer to its nearest class prototype as well as to an alternative augmented view of the test image. With these two additional inputs for each test example, the input batch contains three times the number of samples in the original test batch. Following (Khosla et al., 2020), each of these inputs is passed through a small learnable projection layer and the outputs from this layer are used to finally compute the contrastive loss. Let, $A(x)$ be the set of all images except $x$, and $V(x)$ be the different views of $x$ including the closest class prototype to $x$, then the contrastive loss is formulated as:

$$\mathcal{L}_{\text{CL}} = -\sum_{x \in X} \sum_{v \in V(x)} \log \frac{\exp\big(\text{sim}(z_x, z_v)/\tau\big)}{\sum\limits_{a \in A(x)} \exp\big(\text{sim}(z_x, z_a)/\tau\big)}, \tag{6}$$

where $z_x$, $z_v$ and $z_a$ are the normalized projections of the samples $x$, $v$ and $a$ respectively. $\tau$ is the temperature and $\text{sim}(u,v) = u^T v/(\|u\|\|v\|)$ is the cosine similarity. The overall loss function is formed by summing up the two losses $\mathcal{L}_{CL}$ and $\mathcal{L}_{ST}$.

$$\mathcal{L}_{total} = \mathcal{L}_{ST} + \lambda_{CL}\mathcal{L}_{CL} \tag{7}$$

where $\lambda_{CL} \in [0,1]$ is the hyperparameter controlling the weight of $\mathcal{L}_{CL}$. This loss updates the parameters of the student model $\theta$, while the teacher model is updated by ema of the existing teacher and the student models.

**Inference:** During inference, in accordance with (Döbler et al., 2023), a mean prediction of the student and the teacher model outputs is used for classifying the incoming test images.

## 4 Experiments

**Datasets and Metrics Used**: We evaluate DMSE on several benchmark datasets - DomainNet-126 (Saito et al., 2019), ImageNet-C, CIFAR10-C and CIFAR100-C (Hendrycks & Dietterich, 2019). CIFAR10-C, CIFAR100-C and ImageNet-C consist of 10, 100, and 1000 classes, respectively. Each of these datasets comprises of 15 different corruptions representing new domains with five severity levels of corruption, while DomainNet consists of images from 4 different domains. The sequence of corruptions used for evaluation follows standard practice (Chakrabarty et al., 2023; Döbler et al., 2023; Wang et al., 2022) and we report the error rates on various domains arriving sequentially as well as the average error over all corruptions for the highest severity level (5). For CIFAR10-C and CIFAR100-C, there are 10,000 images per corruption type, while the ImageNet-C split which most previous works (Wang et al., 2022; Niu et al., 2022; Döbler et al., 2023) adopt from RobustBench (Croce et al., 2021) comprises 5,000 images per corruption by default (referred as ImageNet-C-5k) [1]. Inspired by (Press et al., 2023; Chakrabarty et al., 2023), to further investigate the adaptation performance on larger dataset splits, we test our approach on the complete ImageNet-C test set, which comprises 50,000 images per corruption (referred to as ImageNet-C-50k) and we also test on

---

[1]The default ImageNet-C split from RobustBench, as used by most previous baselines, uses 5000 samples per corruption type.

| | Time | $t \longrightarrow$ | | | | | | | | | | | | | | | |
|---|---|---|---|---|---|---|---|---|---|---|---|---|---|---|---|---|---|
| | Method | gaussian | shot | impulse | defocus | glass | motion | zoom | snow | frost | fog | brightness | contrast | elastic | pixelate | jpeg | Mean |
| ImageNet-C-5k | RMT | 80.2 | 76.4 | 74.5 | 77.1 | 74.4 | 66.2 | 57.6 | 57.0 | 59.1 | 48.0 | 39.1 | 60.6 | 47.3 | 42.5 | 43.4 | 60.2 |
| | SANTA | 74.1 | 72.9 | 71.6 | 75.7 | 74.1 | 64.2 | 55.5 | 55.6 | 62.9 | 46.6 | 36.1 | 69.9 | 50.6 | 44.3 | 48.5 | 60.1 |
| | DMSE$^s$ | 78.9 | 72.2 | 71.2 | 72.2 | 70.1 | 62.9 | 55.1 | 53.8 | 57.8 | 45.3 | 35.2 | 63.9 | 45.8 | 41.3 | 43.8 | **58.0** |
| | Source only | 97.8 | 97.1 | 98.2 | 81.7 | 89.8 | 85.2 | 78.0 | 83.5 | 77.1 | 75.9 | 41.3 | 94.5 | 82.5 | 79.3 | 68.6 | 82.0 |
| | BN + Adapt | 85.0 | 83.7 | 85.0 | 84.7 | 84.3 | 73.7 | 61.2 | 66.0 | 68.2 | 52.1 | 34.9 | 82.7 | 55.9 | 51.3 | 59.8 | 68.6 |
| | TENT-cont. | 81.6 | 74.6 | 72.7 | 77.6 | 73.8 | 65.5 | 55.3 | 61.6 | 63.0 | 51.7 | 38.2 | 72.1 | 50.8 | 47.4 | 53.3 | 62.6 |
| | DeYO-cont. | 74.5 | 65.4 | 64.9 | 73.7 | 70.2 | 65.0 | 57.4 | 62.2 | 62.3 | 51.9 | 39.5 | 63.0 | 50.3 | 46.3 | 48.9 | 59.7 |
| | CoTTA | 84.7 | 82.1 | 80.6 | 81.3 | 79.0 | 68.6 | 57.5 | 60.5 | | 48.3 | 36.6 | 66.1 | 47.2 | 41.2 | 46.0 | 62.7 |
| | RDumb | 75.2 | 67.0 | 65.3 | 74.0 | 69.6 | 65.0 | 57.3 | 62.9 | 62.2 | 53.7 | 41.1 | 64.1 | 52.2 | 43.8 | 49.3 | 60.2 |
| | RMT* | 80.3 | 76.9 | 74.0 | 75.6 | 73.8 | 64.8 | 56.6 | 56.6 | 58.2 | 48.3 | 39.6 | 57.8 | 46.6 | 43.2 | 44.4 | 59.8 |
| | SANTA* | 75.3 | 73.2 | 71.5 | 75.5 | 74.6 | 66.0 | 55.7 | 56.3 | 63.0 | 46.6 | 36.9 | 69.4 | 50.1 | 45.3 | 48.4 | 60.5 |
| | DMSE | 79.0 | 72.4 | 70.7 | 72.2 | 70.6 | 63.5 | 55.6 | 54.3 | 57.3 | 45.4 | 35.3 | 64.2 | 46.1 | 41.0 | 44.5 | **58.1** |
| CIFAR10-C | RMT | 24.5 | 20.0 | 25.5 | 13.9 | 24.6 | 14.9 | 13.3 | 16.0 | 15.8 | 15.6 | 11.1 | 15.0 | 18.3 | 14.6 | 16.9 | 17.3 |
| | SANTA | 23.9 | 20.1 | 28.0 | 11.6 | 27.4 | 12.6 | 10.2 | 14.1 | 13.2 | 12.2 | 7.4 | 10.3 | 19.1 | 13.3 | 18.5 | **16.1** |
| | DMSE$^s$ | 24.3 | 21.4 | 26.3 | 11.9 | 25.3 | 12.3 | 10.2 | 14.5 | 14.2 | 11.9 | 7.5 | 10.7 | 17.8 | 14.1 | 19.5 | **16.1** |
| | Source only | 72.3 | 65.7 | 72.9 | 46.9 | 54.3 | 34.8 | 42.0 | 25.1 | 41.3 | 26.0 | 9.3 | 46.7 | 26.6 | 58.5 | 30.3 | 43.5 |
| | BN + Adapt | 28.1 | 26.1 | 36.3 | 12.8 | 35.3 | 14.2 | 12.1 | 17.3 | 17.4 | 15.3 | 8.4 | 12.6 | 23.8 | 19.7 | 27.3 | 20.4 |
| | TENT-cont. | 24.8 | 20.6 | 28.6 | 14.4 | 31.1 | 16.5 | 14.1 | 19.1 | 18.6 | 18.6 | 12.2 | 20.3 | 25.7 | 20.8 | 24.9 | 20.7 |
| | DeYO-cont. | 24.9 | 19.5 | 28.9 | 12.6 | 30.7 | 14.6 | 12.5 | 17.2 | 16.5 | 16.4 | 9.7 | 12.4 | 24.4 | 18.8 | 24.6 | 18.9 |
| | CoTTA | 24.3 | 21.3 | 26.6 | 11.6 | 27.6 | 12.2 | 10.3 | 14.8 | 14.1 | 12.4 | 7.5 | 10.6 | 18.3 | 13.4 | 17.3 | **16.2** |
| | RDumb | 24.3 | 19.2 | 27.7 | 12.7 | 29.1 | 13.9 | 11.5 | 16.2 | 15.3 | 14.8 | 9.3 | 12.9 | 21.5 | 16.2 | 20.6 | 17.6 |
| | RMT* | 24.4 | 20.2 | 25.5 | 12.6 | 25.5 | 14.3 | 12.5 | 15.3 | 15.2 | 14.3 | 10.5 | 13.6 | 17.7 | 13.6 | 16.1 | 16.7 |
| | SANTA* | 24.0 | 19.5 | 28.0 | 11.5 | 28.3 | 12.4 | 10.1 | 14.7 | 14.0 | 12.3 | 7.6 | 10.4 | 19.5 | 14.6 | 20.9 | 16.5 |
| | DMSE | 24.2 | 21.3 | 27.5 | 11.6 | 27.5 | 12.4 | 10.2 | 14.6 | 14.3 | 12.0 | 7.4 | 10.9 | 18.3 | 14.2 | 20.3 | 16.4 |
| CIFAR100-C | RMT | 40.5 | 36.1 | 36.3 | 27.7 | 33.9 | 28.5 | 26.4 | 29.0 | 29.0 | 32.5 | 25.1 | 27.4 | 28.2 | 26.3 | 29.3 | 30.4 |
| | SANTA | 36.5 | 33.1 | 35.1 | 25.9 | 34.9 | 27.7 | 25.4 | 29.5 | 29.9 | 33.1 | 23.6 | 26.7 | 31.9 | 27.5 | 35.2 | 30.3 |
| | DMSE$^s$ | 39.5 | 36.0 | 36.1 | 28.4 | 33.5 | 28.3 | 26.3 | 28.6 | 29.0 | 31.0 | 24.3 | 26.3 | 28.0 | 26.4 | 30.0 | **30.1** |
| | Source only | 73.0 | 68.0 | 39.4 | 29.3 | 54.1 | 30.8 | 28.8 | 39.5 | 45.8 | 50.3 | 29.5 | 55.1 | 37.2 | 74.7 | 41.2 | 46.4 |
| | BN + Adapt | 42.1 | 40.7 | 42.7 | 27.6 | 41.9 | 29.7 | 27.9 | 34.9 | 35.0 | 41.5 | 26.5 | 30.3 | 35.7 | 32.9 | 41.2 | 35.4 |
| | TENT-cont. | 37.2 | 35.8 | 41.7 | 37.9 | 51.2 | 48.3 | 48.5 | 58.4 | 63.7 | 71.1 | 70.4 | 82.3 | 88.0 | 88.5 | 90.4 | 60.9 |
| | DeYO-cont. | 36.4 | 32.8 | 35.8 | 28.7 | 37.7 | 30.8 | 28.4 | 34.1 | 33.0 | 37.1 | 30.0 | 31.3 | 36.3 | 32.5 | 40.2 | 33.7 |
| | CoTTA | 40.1 | 37.7 | 39.7 | 26.9 | 38.0 | 27.9 | 26.4 | 32.8 | 31.8 | 40.3 | 24.7 | 26.9 | 32.5 | 28.3 | 33.5 | 32.5 |
| | RDumb | 37.1 | 34.6 | 39.7 | 34.1 | 44.3 | 39.2 | 38.0 | 44.6 | 45.5 | 50.1 | 45.8 | 53.0 | 57.8 | 54.9 | 62.6 | 45.1 |
| | RMT* | 40.6 | 36.7 | 36.8 | 28.2 | 33.9 | 28.4 | 26.7 | 29.5 | 28.9 | 31.4 | 25.3 | 27.4 | 28.3 | 26.8 | 29.6 | 30.6 |
| | SANTA* | 36.7 | 33.4 | 35.4 | 25.9 | 35.8 | 28.1 | 24.9 | 29.8 | 29.9 | 33.8 | 23.4 | 26.6 | 31.2 | 27.8 | 35.5 | 30.5 |
| | DMSE | 40.0 | 35.9 | 36.9 | 28.3 | 33.4 | 28.4 | 26.2 | 28.7 | 29.3 | 32.2 | 24.4 | 26.5 | 27.9 | 27.1 | 30.8 | **30.4** |
| ImageNet-C-50k | RMT | 73.6 | 65.9 | 64.3 | 74.3 | 72.0 | 71.0 | 69.9 | 70.2 | 71.9 | 70.3 | 66.2 | 74.7 | 68.5 | 67.3 | 67.9 | 69.9 |
| | SANTA | 73.6 | 75.1 | 73.2 | 76.2 | 76.8 | 64.1 | 53.5 | 55.8 | 61.7 | 43.7 | 34.5 | 72.7 | 49.2 | 43.9 | 50.2 | 60.3 |
| | DMSE$^s$ | 73.8 | 69.7 | 69.1 | 72.0 | 71.2 | 61.0 | 52.8 | 55.2 | 58.2 | 44.3 | 34.0 | 65.3 | 46.8 | 41.8 | 48.6 | **57.6** |
| | Source only | 97.8 | 97.1 | 98.1 | 82.1 | 90.2 | 85.2 | 77.5 | 83.1 | 76.7 | 75.6 | 41.1 | 94.6 | 83.0 | 79.4 | 68.4 | 82.0 |
| | BN + Adapt | 84.9 | 84.0 | 84.2 | 85.0 | 84.7 | 73.6 | 61.2 | 65.6 | 66.9 | 52.0 | 34.7 | 83.2 | 55.8 | 51.0 | 60.2 | 68.5 |
| | TENT-cont. | 71.5 | 66.1 | 69.3 | 82.3 | 90.0 | 94.9 | 97.0 | 98.8 | 99.3 | 99.2 | 99.2 | 99.6 | 99.4 | 99.4 | 99.4 | 91.0 |
| | DeYO-cont. | 64.5 | 59.6 | 63.5 | 80.9 | 97.3 | 99.7 | 99.8 | 99.8 | 99.9 | 99.3 | 99.8 | 99.8 | 99.9 | 99.9 | 99.9 | 90.9 |
| | CoTTA | 78.4 | 68.4 | 64.4 | 74.8 | 71.8 | 69.3 | 67.4 | 72.1 | 71.1 | 67.0 | 62.2 | 73.5 | 69.4 | 67.1 | 68.6 | 69.7 |
| | RDumb | 64.9 | 62.6 | 64.1 | 69.1 | 65.8 | 53.7 | 48.6 | 51.9 | 54.5 | 40.4 | 33.4 | 57.7 | 44.9 | 39.9 | 45.5 | **53.2** |
| | RMT* | 74.0 | 66.2 | 64.6 | 74.4 | 72.0 | 71.2 | 69.7 | 70.3 | 71.9 | 70.2 | 65.7 | 74.6 | 68.7 | 66.9 | 67.4 | 69.9 |
| | SANTA* | 74.1 | 74.6 | 73.5 | 76.5 | 76.8 | 63.8 | 53.5 | 55.5 | 61.9 | 43.7 | 34.7 | 73.2 | 49.0 | 43.82 | 50.1 | 60.3 |
| | DMSE | 73.2 | 70.3 | 68.2 | 72.1 | 71.5 | 60.7 | 53.3 | 55.1 | 58.1 | 44.4 | 34.0 | 63.6 | 47.4 | 42.8 | 48.2 | 57.5 |

Table 2: **Classification error rate (%) on CIFAR10-to-CIFAR10-C, ImageNet-to-ImageNet-C, and CIFAR100-to-CIFAR100-C:** Error rates are calculated on the highest corruption severity *i.e.*, level 5. For each dataset, the upper rows list the approaches that use source data for prototyping, while the lower rows list approaches that do not use source data anywhere during adaptation. For RMT and SANTA (which use the source for computing the prototypes by default), we re-implemented them with our proposed source prototype estimation, for fair comparison; for these two methods, superscript $*$ denotes source prototypes are estimated using the pre-trained classifier weights without using original source data (ref. section 3.2). Conversely, for the proposed DMSE, superscript $s$ means source prototypes are obtained by using original source data. Best results are highlighted in **bold**.

DomainNet-126 (Saito et al., 2019), a subset of DomainNet (Peng et al., 2019), comprising ∼18k, ∼30k and ∼24k images in clipart, painting, and sketch, domains respectively. Throughout our experiments, we follow the fully continual TTA setup (Wang et al., 2022; Döbler et al., 2023) wherein there is no assumption of domain switch knowledge being available.

**Implementation Details**: Following existing works (Chakrabarty et al., 2023; Döbler et al., 2023; Wang et al., 2022), we follow the RobustBench (Croce et al., 2021) benchmark and use pre-trained models. The

| #Samples | 2.5k | 5k | 7.5k | 10k | 15k | 25k | 50k |
|----------|------|------|------|------|------|------|------|
| DMSE | 59.5 | 58.1 | 57.9 | 57.8 | 57.7 | 57.6 | 57.5 |
| RDumb | 62.7 | 60.2 | 58.7 | 57.0 | 55.9 | 54.3 | 53.2 |

Table 3: Comparison of trends between DMSE and RDumb on ImageNet-C over different number of images per corruption.

ImageNet-to-ImageNet-C and DomainNet-126 adaptation is performed on a pre-trained ResNet-50 backbone while CIFAR10-to-CIFAR10-C and CIFAR100-to-CIFAR100-C adaptations are performed on WideResNet-28 (Zagoruyko & Komodakis, 2016) and ResNeXt-29 (Xie et al., 2017) respectively. In line with previous works (Wang et al., 2022; Döbler et al., 2023; Chakrabarty et al., 2023) ImageNet-to-ImageNet-C and DomainNet-126 adaptations are performed using an SGD optimizer with lr 0.01, while for CIFAR10-to-CIFAR10-C and CIFAR100-to-CIFAR100-C, an Adam Optimizer with lr 0.001 is used. $\alpha_{min}, \lambda$ and $b_{min}$ are set to $0.99, 0.01$ and $0.2$ respectively for all the datasets. Likewise, the distance threshold $\gamma$ is set to $0.3$ for all the datasets. Following (Wang et al., 2022; Döbler et al., 2023), the hyperparameters have been chosen by performing a small-scale sensitivity analysis on ImageNet-to-ImageNet-C (ref. Supplementary Materials) and the same set is used across all the datasets subsequently. All experiments were conducted on a 24GB NVIDIA A5000 GPU.

## 4.1 Comparison on Benchmark Datasets

We compared against several source-free approaches *e.g.*, CoTTA (Wang et al., 2022) RDumb (Press et al., 2023), Tent (Wang et al., 2021), DeYO (Lee et al., 2024) as well as the source-only baseline, which is a source pre-trained model without any adaptation. DMSE is also compared with RMT (Döbler et al., 2023) and SANTA (Chakrabarty et al., 2023) which require source data to compute class-wise prototypes at the start. A notable strength of our approach is its ability to achieve superior performance without accessing the source data at any stage of the adaptation. However, when provided with source domain data for accurate source prototype estimation (rows denoted with superscript $s$ in Table 2), our model's performance is further enhanced showing its versatility.

For a fair comparison, we ran source-free versions of RMT and SANTA as well, where source prototypes were estimated from the pre-trained classifier only, without using original source data. Our approach consistently achieves superior performance in both source-free and non-source-free setups compared to existing methods. We also experimented with two recent test-time adaptation approaches – Tent (Wang et al., 2021) and DeYO (Lee et al., 2024) run in a CTTA setting (referred to as Tent-cont. and DeYO-cont. respectively). These methods are adapted during test-time by minimizing their own prediction entropy. While such strategies have worked for test-time adaptation, it can not handle continually changing domains at test-time. It can be noted that the performances of the closest approaches RMT and SANTA deteriorate over time in comparison to DMSE, as observed from the error margins for the latter corruptions, in Table 2 and Table 1, across all datasets. This verifies our approach to be more effective in combating catastrophic forgetting and error accumulation.

**Comparison with RDumb**: We extensively compare DMSE with RDumb (Press et al., 2023), a work which challenges the evolution in CTTA techniques. The results from Table 2 and Table 1 show that RDumb particularly performs very well on ImageNet-C-50k. To investigate any underlying trends with the amount of test data, we perform a comparison between the performances of DMSE and RDumb in Table 3. These results suggest that RDumb performs well for more data-intensive CTTA settings wherein a large number of samples from each corruption are available while DMSE can quickly adapt to changing distribution without needing too many sample at test-time. The performance at data scarce scenario is more significant as this reflects the methods's performance for difficult cases and the better performance of the proposed approach in this, shows the ability of DMSE for quicker and more generalizable test-time adaptation.

## 4.2 Ablation Studies and Additional Analysis

We perform ablation experiments for each component of our approach and list our findings in Table 5.

**Need for Controlled Adaptation of Teacher**: In this experiment, instead of dynamically updating the momentum $\alpha$ with input batches, we use a fixed value of $\alpha = 0.999$, as is commonly used in literature Döbler et al. (2023); Wang et al. (2022); Brahma & Rai (2023); Yuan et al. (2023), during the EMA update of the

| Avg Error (%) | Source | BN+Adapt | Tent-cont. | DeYO-cont | CoTTA | RDumb | RMT | SANTA | DMSE |
|---|---|---|---|---|---|---|---|---|---|
| ImageNet-C-50k | 82.0 | 68.5 | 84.4±6.3 | 86.9±5.8 | 65.5±3.6 | **53.6±0.3** | 64.2±5.8 | 60.3±0.2 | 57.1±0.6 |
| CIFAR10-C | 24.7 | 14.2 | 24.5 | 20.0 | 11.1 | 11.8 | **10.4** | 10.7 | 10.5 |
| CIFAR100-C | 33.6 | 30.1 | 79.0 | 31.7 | 27.4 | 28.4 | 27.0 | 26.2 | **26.1** |

Table 4: **Top:** Average error (%) over 10 different corruption sequences of the ImageNet-C dataset. **Bottom:** Average error (%) in the continual adaptation setting with gradually varying severities for the CIFAR10-C and CIFAR100-C datasets.

| Class Prototype | CTA | ImageNet-C | Cifar100-C | Cifar10-C |
|---|---|---|---|---|
| Fixed | – | 59.9 | 30.7 | 17.8 |
| Re-calibrated | – | 59.6 | 30.4 | 17.5 |
| Fixed | ✓ | 58.5 | 30.4 | 16.5 |
| Re-calibrated | ✓ | 58.1 | 30.4 | 16.4 |

Table 5: **Component-wise contributions**: Mean error obtained over 15 corruptions. ✓ in CTA denotes $\alpha$ is dynamically updated.

| Source adaptation | Source | RMT | SANTA | DMSE |
|---|---|---|---|---|
| ✗ | 23.7 | 25.3 | 23.6 | 24.4 |
| ✓ | 23.7 | 25.1 | 23.8 | 23.5 |

Table 6: Classification error rates on clean test set of CIFAR100 after performing CTTA on 15 corruption types in the CIFAR100-C.

teacher. Table 5 shows that controlled teacher adaptation (denoted by a checkmark in column CTA) leads to significant performance improvement across all datasets.

**Re-calibrating Class-wise Prototypes**: Class-wise prototypes play a pivotal role in disentangling the target domain features by aligning with them. Previous works Chakrabarty et al. (2023); Döbler et al. (2023) tend to continue with the same class-wise prototypes initially computed from the source data. We propose to re-calibrate the class-wise prototypes with the changing target features as new target data arrives. In this experiment we compare the performance between fixed prototypes and our proposed re-calibration. As shown in Table 5, the reduction in error rates in going from fixed to continually evolving prototypes (ref. column 'Class prototype') is a testament to our hypothesis.

**Performance over different corruption sequences:** Following Chakrabarty et al. (2023), to investigate generalizability, we performed experiments over 10 random permutations of the 15 corruption sequences of the ImageNet-C-50k. Table 4 (top) reports the mean error over these permutations. Since the 10 sequences used are randomly sampled orderings of the corruptions, the results too have some variability and may alter the relative performance across methods. DMSE achieves the best results on CIFAR100-C, while ranking the second best in other cases. The consistently better performance of DMSE on difference corruption sequences is also shown in Fig 3(a), showcasing its robustness and adaptability to diverse corruption patterns.

**Performance over gradual domain-shifts:** In a standard setting, the corruption types change from one noise to other at the maximum severity level. However, there can be scenarios where the domain changes are more gradual compared to the standard setup. Hence, following Wang et al. (2022), we evaluate our approach in the gradual setup where the severity levels within each noise change gradually as follows:

$$\underbrace{\ldots \to 2 \to 1}_{ct\text{-}1 \text{ and before}} \to \underbrace{1 \to 2 \to 3 \to 4 \to 5 \to 4 \to 3 \to 2 \to 1}_{ct \text{ corruption type, with changing severity}} \to \underbrace{1 \to 2 \to \ldots}_{ct\text{+}1 \text{ and after}}$$

ct represents the corruption type. Table 4 (bottom), presents the performance in the gradual test-time adaptation setup. DMSE performs at par or better than existing approaches.

**Performance trends over varying batch sizes:** As observed in Table 7, a larger batch size results in improved performance across all methods, with the performance across almost all batch sizes being better for DMSE.

**Revisiting the clean source data**: While a CTTA model adapts to changing conditions, it is also important to maintain a good performance on the original source distribution. Following Chakrabarty et al. (2023), we

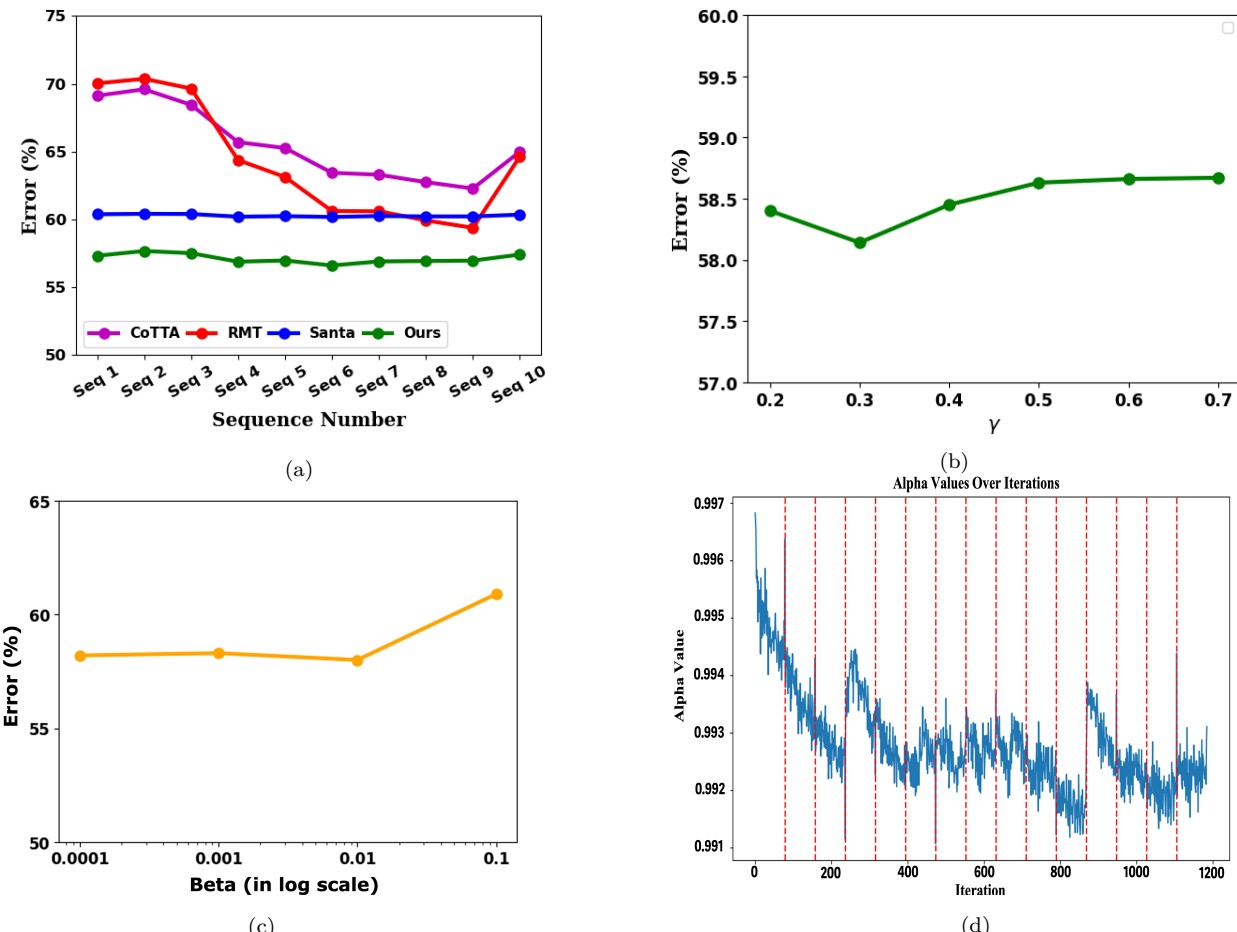

Figure 3: **Ablation studies on `DMSE` over ImageNet-C:** (a) Comparison of mean errors over 10 different sequences of the 15 corruptions, using different CTTA methods (b) Mean errors over 15 corruptions with varying cosine-distance threshold for class-wise prototype estimation (c) Mean errors over 15 corruptions with varying scaling factor ($\beta$). (d) Variation of Momentum ($\alpha$) over 15 corruption types. The red dotted lines indicate the boundaries between different corruption categories

| Batch Sizes | 16 | 32 | 64 | 128 |
|---|---|---|---|---|
| RMT | 84.2 | 60.8 | 59.8 | 59.7 |
| SANTA | 68.1 | 63.1 | 60.5 | 59.3 |
| DMSE | 70.2 | 60.9 | 58.1 | 58.0 |

Table 7: Classification error rates of different batch sizes for CTTA on ImageNet-C-5k.

used the model adapted on CIFAR100-C to perform inference on a held-out test data of clean CIFAR100 (ref. Table 6). The percentage error of the original source pre-trained model is 23.7. The top row shows the performance in this setting where SANTA performs best with even lower error compared to the original source pre-trained model. However, CTTA enables us to adapt in test time and thus, it is quite natural to exploit this ability on the held-out source data in test time. Allowing the approaches to continue adapting to the source test data shows the superiority of our model over the competing approaches (bottom row). It is worth noting that both SANTA and RMT uses source data for accurately estimating the class prototypes during adaptation which is not required in our case.

**Sensitivity analysis on** $\gamma$: We ran a sensitivity analysis of the threshold $\gamma$ used to update the class-wise prototypes (ref Eqn. 3). Figure 3(b) shows the analysis on ImageNet-C dataset. The best performance is obtained with $\gamma = 0.3$ and this value is used throughout for our experiments.

**Sensitivity Analysis on** $\alpha$**,** $\beta$**, and** $e$: As described in Section 4, we perform a small-scale sensitivity analysis on the ImageNet-C dataset to determine the optimal hyperparameters and use them across all datasets. Table 8 and Figure 3(c) show the results obtained from the experiments conducted over a grid search for hyperparameters in Eqn. 2. Based on these results, $\alpha_{min}, \beta$ and $e_{min}$ are set to $0.99, 0.01$ and $0.2$ respectively for all the datasets.

| $\alpha_{min}$ \ $e$ | 0.1 | 0.15 | 0.2 | 0.25 | 0.3 |
|---|---|---|---|---|---|
| 0.98 | 60.8 | 59.1 | 58.6 | 58.9 | 59.6 |
| 0.985 | 60.1 | 58.5 | 58.1 | 58.8 | 59.7 |
| 0.99 | 58.8 | 58.3 | 58.1 | 58.8 | 59.6 |
| 0.995 | 59.8 | 59.8 | 59.4 | 59.5 | 60.7 |
| 0.999 | 60.9 | 60.6 | 59.9 | 60.4 | 60.9 |

Table 8: **Sensitivity Analysis of** $\alpha_{min}$ **and** $e$: Mean error obtained over 15 corruptions on ImageNet-C 5k dataset.

**Deep-dive into dynamic momentum adjustment**: Fig. 3(d) depicts the variation of teacher momentum over a sequence of changing corruptions as observed during continual test-time adaptation. As is clear from the trend, the teacher model's momentum value shows a tendency to decrease over the sequence of corruptions suggesting more imbibition from student with increasing student prediction confidence i.e. lower batch entropy. While this imbibition is desirable, we also want to prevent too much drift of the teacher away from the original target distribution, since corruptions coming consecutively might be dissimilar from each other but will still hold a certain degree of resemblance with the original target distribution. For this purpose, we reset our teacher model intermediately if too much drift is observed, as can be observed by the intermediate spikes in teacher momentum to withhold too much knowledge imbibition from student model.

**Different ways of updating the prototypes**: As described in Eqn. 3, We tried with both initial ($p_0^c$) as well as immediately previous prototypes ($p_{t-1}^c$) for getting $p_t^c$, and observed a better performance in the former. The comparatively higher error rate in the latter setting with changing prototype centers, as seen in Table 9, could possibly be attributed to more than desired drift of the prototypes from the real prototypes.

| $p_t^c$ using | Error Rate |
|---|---|
| $p_0^c$ | 58.1 |
| $p_{t-1}^c$ | 59.1 |

Table 9: **Comparison between class prototype updation methods:** The mean errors over 15 corruptions on ImageNet-C-5k shows that updates based on the initial source prototypes ($p_0^c$) work better compared to updates based on the immediately previous prototypes. This is possibly due to increased drift from the actual source class features in the later case.

## 5 Limitations

As shown in Table 6, `DMSE` underperforms compared to SANTA (Chakrabarty et al., 2023) and the source pretrained model in maintaining good inference performance on the held-out test data of clean CIFAR-100 when adapted on CIFAR-100. This reduced performance relative to other approaches can be attributed to the dynamic momentum mechanism, which increases adaptability but also leads to more drift. However, when adaptation is allowed on the held-out test data of the source domain, our model outperforms the competing approaches, demonstrating its ability to quickly adapt to new domains.

## 6 Conclusion

In this paper, we addressed the challenge of continual test-time adaptation with our proposed (`DMSE`) approach. `DMSE` enhances the model performance across evolving target domains by using a controlled mean teacher updated using dynamically decided momentum. We also estimate class-wise source prototypes directly from the pre-trained source model. This method mitigates error accumulation and ensures robust adaptation without requiring access to source data at any stage of the pipeline addressing data storage and privacy constraints. We demonstrate the effectiveness of our approach on four benchmark datasets, significantly outperforming several competing methods, some of which require access to source data or its statistics to warmup the process.

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

# A  Appendix

## A.1  Effect of momentum on different corruptions

We investigate the effect of momentum ($\alpha$), in a vanilla mean teacher-student setup, on different corruptions by adapting using an RMT-like approach on one corruption at a time. Fig. 4 shows how the error rates change for different domains of the ImageNet-C 5k dataset for different momentum ($\alpha$) values, which reinforces our motivation that having a high momentum throughout isn't optimal when adapting over long sequences of continually changing domains.

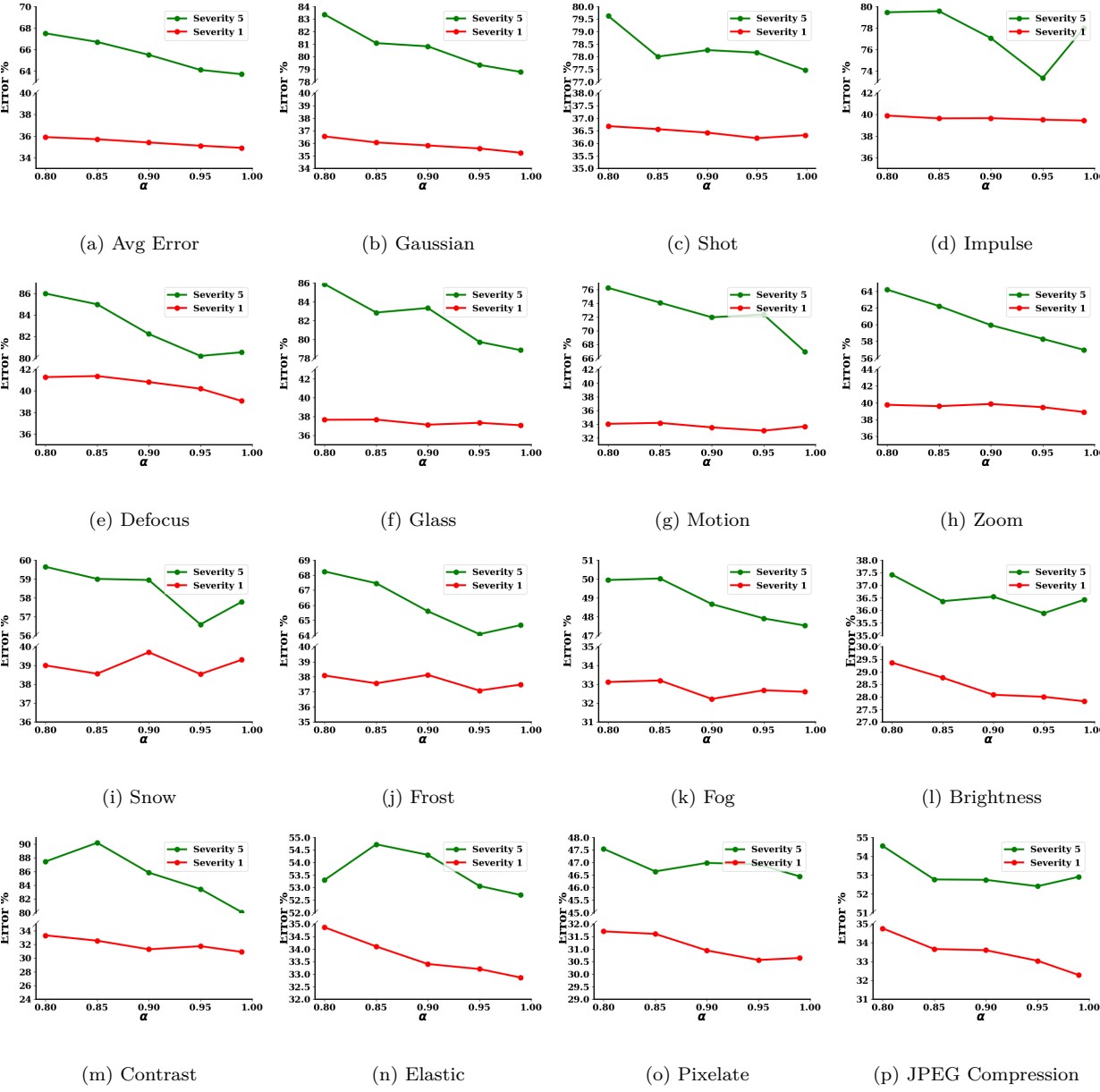

Figure 4: **Motivation for dynamic momentum:** (a) The mean of all the single noise adaptation errors over the 15 corruptions in ImageNet-C-5k. (b)-(p) The errors obtained on different corruption domains of ImageNet-C-5k, taken one at a time. We calculated the error rates for different $\alpha$ values over different types of noises and different severity levels of the Imagenet-C dataset. We found that while on average a high $\alpha$ helps improve the average accuracy, as seen in (a), different types of noises perform optimally at different $\alpha$ values as seen in (b)-(p), thus justifying a need for a dynamic momentum adjustment. Additionally, these optimum $\alpha$ values also vary with varying noise severities.

## A.2 Sensitivity Analysis of $\tau$ and $\lambda_{CL}$

Following the design choices in RMT (Döbler et al., 2023), we set the contrastive loss temperature to 0.1 and its weight to 0.5. To further verify the robustness of this setting, we performed a sensitivity analysis over the ImageNet-C-5k benchmark by varying both the temperature and the loss weight. As shown in Table 10, our results show that the original values consistently provide the best trade-off between adaptation performance and stability across corruption types.

| $\tau$ \ $\lambda_{CL}$ | 0 | 0.25 | 0.5 | 0.0.75 | 1.0 |
|---|---|---|---|---|---|
| 0.1 | 60.97 | 59.17 | **58.1** | 58.58 | 58.27 |
| 0.5 | 59.34 | 59.14 | 59.15 | 59.11 | 59.14 |
| 1.0 | 59.59 | 59.63 | 59.75 | 59.28 | 58.96 |

Table 10: **Sensitivity Analysis of $\tau$ and $\lambda_{CL}$:** Mean error obtained over 15 corruptions on ImageNet-C-5k dataset.

## A.3 Need for Resetting the Teacher Model

We perform an experiment to evaluate the effectiveness and need of the teacher model resetting as mentioned in Section 3.1. Performing continual adaptation without resetting gives a poorer average error rate of 59.9% on the ImageNet-C-5k dataset as compared to the result of 58.1% obtained via DMSE with the resetting technique active.

## A.4 Protection Against Teacher Collapse

To analyse the effectiveness of our approach in providing protection against teacher collapse over long streams, we conducted additional experiments on the ImageNet-C-50k dataset, which contains 50,000 images per corruption domain. We evaluated our method using both fixed and dynamic momentum settings across these long sequences. The fixed momentum approach gives a 70.2% error rate while our approach with dynamic momentum gives an error rate of 57.5% on this ImageNet-C-50k sequence. This significant improvement demonstrates that dynamic momentum not only enhances adaptability over long streams but also provides effective protection against error accumulation and teacher collapse, thereby highlighting the robustness of DMSE in continual adaptation scenarios.

## A.5 Importance of the Projection Layer in Contrastive Loss

Contrastive loss helps align the test feature distribution with the source domain, where the pre-trained model is more reliable and well-calibrated. This alignment enhances the model's generalization capability in the target domain. Adding a projection layer significantly improves the performance as shown (Bachman et al., 2019; Chen et al., 2020b). Following the best practices as detailed in (Appalaraju et al., 2020) non-linear projection layer helps preserve only the most discriminative information to make classification. We have also performed an experiments with and without the projection layer to reinvestigate the same empirically. Without the projection layer we get an error rate of 60.6% on the ImageNet-C-5k dataset compared to the 58.1% as obtained using DMSE.

## A.6 Validation of Source Estimation and Prototype Alignment

To validate the core design of our prototype-based approach, we provide a t-SNE visualization demonstrating the alignment between classifier-derived prototypes and true source prototypes, as well as the proximity of test-time prototypes to them. Specifically, we randomly sampled 500 test-time prototypes $(p_t^c)$ generated

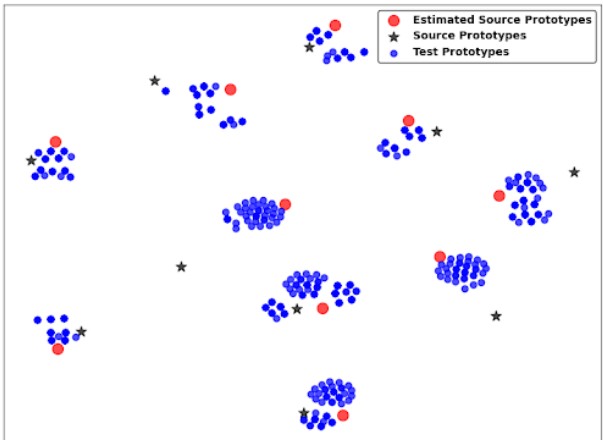

Figure 5: **Validation of Prototype Alignment:** t-SNE plot showing 500 test-time prototypes (blue), source prototypes from classifier weights (red), and source prototypes from source data (black). Prototypes estimated using classifier weights align with true source prototypes, and test-time prototypes remain close to them.

| Loss Type | ImageNet-C-5k | Cifar100-C | Cifar10-C |
|-----------|---------------|------------|-----------|
| CE | 62.9 | 33.4 | 23.6 |
| SCE | 58.1 | 30.4 | 16.4 |

Table 11: **Ablation study comparing symmetric cross-entropy (SCE) loss with categorical cross-entropy (CE) loss on ImageNet-C-5k, CIFAR10-C, and CIFAR100-C.** Performance consistently drops when replacing SCE with CE, demonstrating the effectiveness of SCE in the DMSE framework.

at different time steps during the continual test-time adaptation process from the CIFAR10-C dataset and plotted their t-SNE representations (blue dots). Here, the estimated source prototypes (red dots) correspond to classifier weights from the WideResNet-28 backbone as used during CIFAR10 to CIFAR10-C continual test-time adaptation, and these are the same as our initial prototype estimates ($p_0^c$ as used in Eqn. 3). The original source prototypes (black stars) correspond to the mean of features obtained by passing the CIFAR10 source domain data through the same pretrained feature extractor (i.e. the entire model without the classifier layer) from the WideResNet-28 backbone, and prior works like RMT (Döbler et al., 2023) or SANTA (Chakrabarty et al., 2023) use this method to obtain source prototypes. Alongside, we plotted the source prototypes estimated from the classifier weights (red dots) and those computed directly from the source domain data (black stars).

As shown in Fig. 5, we obtain 10 clusters depicting 10 different classes of the CIFAR10 dataset, and the estimated source prototypes, derived from the classifier weights, are observed to be more or less well aligned with the true source prototypes obtained from the source data. Moreover, the test-time prototypes remain consistently close to these source prototypes, validating the stability and reliability of our prototype estimation throughout the adaptation process. This visualization reinforces the consistency and alignment of different types of prototypes, which along with the experimental results from Table 5 supports the effectiveness of our prototype-based formulation.

## A.7 Significance of SCE loss

To assess the effectiveness of the symmetric cross-entropy (SCE) loss within our DMSE framework, we performed an ablation study in which SCE was replaced with the standard categorical cross-entropy (CE) loss. We conducted experiments on three benchmark datasets – ImageNet-C-5k, CIFAR10-C, and CIFAR100-C. As shown in Table 11, substituting CE for SCE consistently led to a decrease in performance across all datasets. These results confirm that the choice of SCE loss is critical to achieving the reported performance gains.

