# OpenReview forum: "Source-Free Controlled Adaptation of Teachers for Continual Test-Time Adaptation"
_TMLR — Rejected by TMLR_

### Review · Reviewer_Acqs · 2025-06-18

**Summary Of Contributions:**

This paper proposes DMSE, a source-free continual test-time adaptation framework that dynamically controls teacher adaptation and leverages class prototypes. The controlled teacher adaptation adjusts the exponential moving average coefficient based on the batch-average entropy, where student models transfer more knowledge with lower batch entropy. Also, DMSE extracts class-wise prototypes directly from the pre-trained classifier weights and uses them as an anchor for updating the test-time sample prototypes. DMSE combines symmetric cross-entropy loss and contrastive loss as a final objective. DMSE shows performance improvements over source-free prototype-based baselines.

**Audience:**

Yes

**Broader Impact Concerns:**

There are no significant broader impact concerns compared to recent advances in test-time adaptation.

**Claims And Evidence:**

Yes

**Requested Changes:**

1. Please explain why the lower $\alpha$ for lower entropy is better for the adaptation. If the batch has a higher entropy due to distribution shift, it implies high uncertainty; in such cases, does it make sense to trust the student less and update the teacher less (i.e., use higher $\alpha$)? Please clarify why lower entropy justifies a greater update from the student. Also, how does this align with the observations in Figure 1?

1. Please explain why the prototype-based training is essential in the teacher-student TTA framework.

1. Please explain that the initial prototype ($p^c$) is a reasonable estimate of the class prototype.

1. Please add a theoretical/empirical explanation that the shifted prototype (of corrupted images) still aligns with the initial prototype (of original images), since Equation 3 uses the distance between the initial prototype and the sample feature as a sample selection metric.

1. The use of symmetric cross-entropy (SCE) loss does not align with the previous explanations on the class prototypes. The paper should further justify the need for SCE loss and how it aligns with the class prototype.

1. The motivation and the detailed implementation of Khosla et al. (2020) are not properly explained. For example, why do we need an additional small learnable projection layer? Why can't we use the original model's projection layer?

1. Please explain the advantage of combining symmetric cross-entropy and contrastive loss.

1. Please consider including ViT experiments and recent teacher-student baselines, including RoTTA [1] and PETAL [2].


Minor comments

1. The paper is poorly formatted. Please add the citations properly, using brackets. The citation Niu et al. (2022) is duplicated.

1. I recommend avoiding including different experiments in the same table (e.g., Table 4).

1. Why is the $\alpha$-scale different in Figures 1(b) and 1(c)?

1. Typo in Figure 1: "orange" distribution -> "red" distribution.

**Strengths And Weaknesses:**

Strengths

1. Dynamic control of teacher adaptation is an underexplored problem in the teacher-student framework in TTA.

1. The method introduces a class-prototype-based approach for sample selection and adaptation, providing an interpretable way to guide continual adaptation over time.

1. The paper provides extensive experiments on various datasets, baselines, and scenarios.



Weaknesses

1. Lack of justification for the controlled teacher adaptation: why is it better to assign a lower $\alpha$ when average batch entropy is lower?

1. Lack of justification for the class-wise prototype estimation: (1) need for prototype-based training, (2) method for getting an initial prototype, and (3) the alignment between the initial and test-time prototypes.

1. Lack of justification for the loss design: (1) The role of symmetric cross-entropy is unclear, considering the contributions of class prototypes. (2) Insufficient explanation of the Khosla et al. (2020) framework. (3) The advantage of combining two losses is unclear.

1. The experimental setting needs improvement. (1) ImageNet-C-5k should not be included in the main results, as it is not a standard setting, and EATA (as cited in the paper) does not use 5k samples. (2) Table 1 does not include the full baselines. (3) The paper lacks recent teacher-student baselines, including RoTTA [1] and PETAL [2]. (4) The paper does not explore other model architectures, such as Vision Transformer (ViT) models.

1. DMSE does not outperform simpler or more recent TTA baselines (e.g., CoTTA, RDumb), which questions the practical advantage of the proposed method.






[1] Longhui Yuan, Binhui Xie, and Shuang Li. Robust test-time adaptation in dynamic scenarios.
In Proceedings of the IEEE/CVF Conference on Computer Vision and Pattern Recognition, pp.
15922–15932, 2023.

[2] Dhanajit Brahma and Piyush Rai. A probabilistic framework for lifelong test-time adaptation.
In Proceedings of the IEEE/CVF Conference on Computer Vision and Pattern Recognition, pp.
3582–3591, 2023.

---

> ### Author Response · Authors · 2025-07-13
> **Response to Reviewer Acqs (Part 1/4)**
>
> We thank the reviewer for recognising the importance of dynamic teacher adaptation in TTA and for appreciating our class-prototype-based approach as an interpretable solution. We also sincerely appreciate the positive remarks on our experimental rigor across diverse datasets and settings.  Below are our responses to the specific concerns.
>
> > (Point 1 in Weaknesses and Requested changes) Please explain why the lower $\alpha$ for lower entropy is better for the adaptation. If the batch has a higher entropy due to distribution shift, it implies high uncertainty; in such cases, does it make sense to trust the student less and update the teacher less (i.e., use higher )? Please clarify why lower entropy justifies a greater update from the student. Also, how does this align with the observations in Figure 1?
>
> When the student model produces predictions with higher entropy it signals considerable uncertainty in the prediction from a possibly noisy test batch, conversely, lower entropy indicates confident predictions from the student model. This correlation in the prediction correctness and entropy has been indicated in Fig. 1.a. Due to this, when the student model makes confident predictions on a test-batch (i.e. lower entropy), a lower $\alpha$ allows the teacher model to imbibe more information from the confident student predictions (ref. Eqn. (1)). As observed in Fig. 1.c., for different corruptions optimal error rates are achieved at different $\alpha$ values. Hence, as elaborated in Section 3.1, the combination of observations from Fig. 1a-c motivate our adoption of dynamic $\alpha$ based on test-batch entropies.
> > (Point 2 in Weaknesses and Requested changes) Please explain why the prototype-based training is essential in the teacher-student TTA framework.
>
> In the fully source-free test-time domain adaptation setting, we cannot assume access to any source or original target domain data. This constraint directed us to rely solely on the pre-trained model itself. So, we extract source prototypes directly from the classifier layer of the pretrained model, which act as stand-ins for the original domain data.
> These prototypes serve several crucial roles:
> - By leveraging the classifier weights as class prototypes, we maintain a reference to the clean source domain distribution. This helps prevent the model from drifting excessively as it encounters new and continually changing target domains.
> - As the model adapts to new domains, the prototypes are dynamically updated using confidently pseudo-labeled target samples. This continual recalibration ensures that the prototypes remain relevant and effective for aligning features from evolving target distributions.
> - Prototype-based training allows the model to derive meaningful source information without the presence of any original source domain data. Our ablation studies (ref. Table 5) demonstrate that incorporating and recalibrating prototypes leads to significant improvements in adaptation performance, all without requiring access to the original data.
>
> > (Point 2 in Weaknesses and Point 3 in Requested changes) Please explain that the initial prototype ($p^c$) is a reasonable estimate of the class prototype.
>
> The initial prototypes, $p^c$, are extracted from the classifier layer of the pretrained model. The model can be thought of as a composition of a feature extractor ($g$) and linear classifier ($h$). The output logits are obtained as $y= h(g(x)) = W_hg_x$ and thus each element in $y$ is a result of the dot product between a row (vector) of the weight matrix $W_h$ and the feature vector $g_x$, and for an image belonging to the $c$-th class, the dot product value would be highest for the $c$-th row of $W_h$. This suggests that features from images belonging to the $c$-th class has a high similarity with the $c$-th row vector from $W_h$, thus making these vectors a good candidate for the initial class-wise prototypes, **in absence of source data**. This is written more elaborately in Section 3.2 and also in ablation experiments Table 5.
>
> > (Point 2 in Weaknesses and Point 4 in Requested changes) Please add a theoretical/empirical explanation that the shifted prototype (of corrupted images) still aligns with the initial prototype (of original images), since Equation 3 uses the distance between the initial prototype and the sample feature as a sample selection metric.
>
> The prototype updates in our approach are performed by aggregating the features of test samples that are already close to the initial prototype centers, as controlled by the threshold $\gamma$ in Eqn 3. Consequently, the dynamically updated prototypes obtained from Eqn 3, remain well-aligned with the initial prototypes throughout the adaptation process. This alignment is maintained because only features within a certain proximity to the original prototypes contribute to the update, ensuring consistency over time.

---

> ### Author Response · Authors · 2025-07-13
> **Response to Reviewer Acqs (Part 2/4)**
>
> > (Point 3 in Weaknesses and Point 5 in Requested changes) The use of symmetric cross-entropy (SCE) loss does not align with the previous explanations on the class prototypes. The paper should further justify the need for SCE loss and how it aligns with the class prototype.
>
> We appreciate the reviewer’s observation. The use of SCE in our work follows RMT [1], where it was shown to improve stability and accuracy in continual test-time adaptation. This choice is not our contribution but a design decision grounded in prior evidence.
> RMT showed that SCE leads to more stable, accurate adaptation across changing domains in continual test-time scenarios. Specifically, in case of standard cross entropy loss, samples with low confidences dominate the overall gradient. This can be obstructive in the setting of self-training, since low confidence samples are typically more inaccurate as shown in our experiments. The reverse cross entropy term in SCE helps tackle this by ensuring the value of the gradient is higher for samples the teacher is more confident about. This is also marked by the experimental results in Table 4 of [1] where the introduction of symmetric cross entropy loss brings about an average improvement of 3% across datasets.
>
> > (Point 3 in Weaknesses and Point 6 in Requested changes) The motivation and the detailed implementation of Khosla et al. (2020) are not properly explained. For example, why do we need an additional small learnable projection layer? Why can't we use the original model's projection layer?
>
> Thank you for the insightful comment. Contrastive loss helps align the test feature distribution with the source domain, where the pre-trained model is more reliable and well-calibrated. This alignment enhances the model’s generalization capability in the target domain. Adding a projection layer significantly improves the performance as shown by [2, 3]. Following the best practices as detailed in  [4] non-linear projection layer helps preserve only the most discriminative information to make classification. We have also performed an experiment at our end with and without the projection layer. Without the projection layer we got an error rate of 60.6% on the ImageNet-C-5k dataset compared to the 58.1% as obtained using DMSE.
>
> > (Point 4 in Weaknesses) ImageNet-C-5k should not be included in the main results, as it is not a standard setting, and EATA (as cited in the paper) does not use 5k samples.
>
> Thank you for your observation. We would like to clarify that previous baselines, including RMT [1], CoTTA [5], and EATA [7], have commonly reported results on the ImageNet-C-5k dataset, as this has been the default configuration in RobustBench. However, later works like SANTA [6], RDumb [8] etc. extended the benchmark by incorporating more images per corruption (50000 images per corruption). To differentiate between these two settings using ImageNet-C dataset, authors in SANTA [6], proposed this notation of ImageNet-C-5k and ImageNet-C-50k. We also followed the same. To ensure transparency, we have updated Section 4 with a footnote to provide this context.
>
> > (Point 5 in Weaknesses) DMSE does not outperform simpler or more recent TTA baselines (e.g., CoTTA, RDumb), which questions the practical advantage of the proposed method
>
> As indicated in Tab. 1 and Tab. 2, our DMSE approach consistently outperforms CoTTA [5] with the difference being even more significant for longer data sequences as observed for ImageNet-C-50k. Below we duplicate the result comparison between these methods as reported in Tab. 1 and 2 in the paper.
> | Method   | CIFAR-10-C | CIFAR-100-C | ImageNet-C-5k | ImageNet-C-50k | DomainNet-126 |
> | -------- | --------   | -------- | -------- |-------- |-------- |
> | CoTTA    | 16.2       | 32.5     | 62.7     | 69.7     | 43.4     |
> | DMSE     | 16.4       | 30.4     | 58.1     | 57.5     | 37.4     |
>
> We also outperform RDumb [8] on most datasets like DomainNet, CIFAR-10C, CIFAR100-C and ImageNet-C-5k. While RDumb does perform well over longer sequences of data, it is highly data-intensive in order to acheive this performance gain as highlighted in Tab 3 and elaborated on in Section 4.1 - Comparison with RDumb. This makes our approach much more practical and applicable in real life scenarios since large amounts of data spread may not always be available during continual adaptation.

---

> > ### Author Response · Authors · 2025-07-13
> > **Response to Reviewer Acqs (Part 3/4)**
> >
> > > (Point 5 in Weaknesses) DMSE does not outperform simpler or more recent TTA baselines (e.g., CoTTA, RDumb), which questions the practical advantage of the proposed method
> >
> > As indicated in Tab. 1 and Tab. 2, our DMSE approach consistently outperforms CoTTA [5] with the difference being even more significant for longer data sequences as observed for ImageNet-C-50k. Below we duplicate the result comparison between these methods as reported in Tab. 1 and 2 in the paper.
> > | Method   | CIFAR-10-C | CIFAR-100-C | ImageNet-C-5k | ImageNet-C-50k | DomainNet-126 |
> > | -------- | --------   | -------- | -------- |-------- |-------- |
> > | CoTTA    | 16.2       | 32.5     | 62.7     | 69.7     | 43.4     |
> > | DMSE     | 16.4       | 30.4     | 58.1     | 57.5     | 37.4     |
> >
> > We also outperform RDumb [8] on most datasets like DomainNet, CIFAR-10C, CIFAR100-C and ImageNet-C-5k. While RDumb does perform well over longer sequences of data, it is highly data-intensive in order to acheive this performance gain as highlighted in Tab 3 and elaborated on in Section 4.1 - Comparison with RDumb. This makes our approach much more practical and applicable in real life scenarios since large amounts of data spread may not always be available during continual adaptation.
> >
> > > (Point 7 in Requested changes) Please explain the advantage of combining symmetric cross-entropy and contrastive loss.
> >
> > Our choice of using symmetric cross-entropy was inspired from RMT which showed that SCE loss leads to more stable and accurate adaptation across changing domains in continual test-time scenarios. We have elaborated this in our response to (Point 3 in Weaknesses and Point 5 in Requested changes). Contrastive loss, on the other hand, plays a crucial role in leveraging augmented views of the same input to enforce representational invariance and enhance feature alignment with the source domain. This alignment enhances the model’s generalization capability in the target domain.
> >
> > > (Point 4 in Weaknesses and Point 8 in Requested changes) Please consider including ViT experiments and recent teacher-student baselines, including RoTTA [1] and PETAL [2].
> >
> > Both RoTTA and PETAL primarily report results on ResNet/ResNeXt architectures. To ensure a fair comparison, we reran RoTTA (using the implementation as per https://github.com/mariodoebler/test-time-adaptation) with both ResNet-50 and ViT backbones on ImageNet-C, obtaining error rates of 67.6% and 58.4%, respectively. Our approach in DMSE achieves a lower error rate of 58.1% on ImageNet-C-5k, even when using the ResNeXt backbone.
> > For PETAL, although a ViT implementation was not readily available, our DMSE technique achieves superior results compared to the reported PETAL performance with ResNet/ResNeXt backbones on CIFAR10-to-CIFAR10C, CIFAR100-to-CIFAR100C, and ImageNet-to-ImageNetC, as shown below.
> >
> >
> > | Method   | CIFAR-10-C | CIFAR-100-C | ImageNet-C-5k |
> > | -------- | --------   | -------- | -------- |
> > | PETAL    | 15.9       | 31.4     | 62.7     |
> > | DMSE     | 16.4       | 30.4     | 58.1     |

---

> > > ### Author Response · Authors · 2025-07-13
> > > **Response to Reviewer Acqs (Part 4/4)**
> > >
> > > >Minor Comments Addressal
> > >
> > > Thank you for pointing out the suggested changes. We have corrected these in the revised submission.
> > >
> > > >I recommend avoiding including different experiments in the same table (e.g., Table 4).
> > >
> > > The experimental results under Table 4 were clubbed due to page limit constraints.
> > >
> > > >Why is the $\alpha$-scale different in Figures 1b and 1c?
> > >
> > > The range of $\alpha$ chosen in Figure 1b is to allow a clear depiction of the trend of changing error rate with $\alpha$ hence this is a wider $\alpha$ range, while in Figure 1c, we show the counts over the best performing $\alpha$ for each corruption type. Owing to this there is a slight difference in the $\alpha$-scale in these 2 figures.
> > >
> > > *References:*
> > > 1. *Döbler, Mario, Robert A. Marsden, and Bin Yang. "Robust mean teacher for continual and gradual test-time adaptation." Proceedings of the IEEE/CVF Conference on Computer Vision and Pattern Recognition. 2023.*
> > > 2. *Bachman, Philip, R. Devon Hjelm, and William Buchwalter. "Learning representations by maximizing mutual information across views." Advances in neural information processing systems 32 (2019).*
> > > 3. *Chen, Ting, et al. "A simple framework for contrastive learning of visual representations." International conference on machine learning. PmLR, 2020.*
> > > 4. *Appalaraju, Srikar, et al. "Towards good practices in self-supervised representation learning." arXiv preprint arXiv:2012.00868 (2020)*
> > > 5. *Qin Wang, Olga Fink, Luc Van Gool, and Dengxin Dai. Continual test-time domain adaptation. In Proceedings of Conference on Computer Vision and Pattern Recognition, 2022.*
> > > 6. *Goirik Chakrabarty, Manogna Sreenivas, and Soma Biswas. SANTA: Source Anchoring Network and Target Alignment for Continual Test Time Adaptation. Transactions on Machine Learning Research, 2023. ISSN 2835-8856.*
> > > 7. *Shuaicheng Niu, Jiaxiang Wu, Yifan Zhang, Yaofo Chen, Shijian Zheng, Peilin Zhao, and Mingkui Tan. Efficient test-time model adaptation without forgetting. In The Internetional Conference on Machine Learning, 2022a.*
> > > 8. *Ori Press, Steffen Schneider, Matthias Kuemmerer, and Matthias Bethge. RDumb: A simple approach that questions our progress in continual test-time adaptation. In Thirty-seventh Conference on Neural Information Processing Systems, 2023.*

---

> > > > ### Author Response · Authors · 2025-08-02
> > > > **Follow-Up on Rebuttal Submission**
> > > >
> > > > Dear reviewer,
> > > >
> > > > We sincerely thank all your efforts and apologize for sending out this note regarding our paper. We have addressed all the thoughtful questions and suggestions raised in the review and have updated the paper/rebuttal. We will be glad to have further discussion to see if our response solves the concerns. Thank you for your service!
> > > >
> > > > Best wishes,
> > > >
> > > > Authors

---

> > > > > ### Comment · Reviewer_Acqs · 2025-08-04
> > > > >
> > > > > I acknowledge the authors' rebuttal and follow-up.
> > > > >
> > > > > Here are my remaining concerns:
> > > > >
> > > > > - There is no validation that classifier weights align with true source prototypes, nor that test-time prototypes remain close to them. This undermines the central prototype-based design.
> > > > > - The benefits of prototype recalibration are marginal (Table 5), so it questions the need for recalibration.
> > > > > - The use of symmetric cross-entropy loss is adopted from prior work without ablation in this context; its effectiveness in DMSE is assumed rather than demonstrated via ablation studies.
> > > > > - DMSE does not consistently outperform recent baselines like RDumb, particularly on longer test-time streams, which questions its generalizability.

---

> ### Author Response · Authors · 2025-08-09
> **Response to Remaining Concerns (Part 1/2)**
>
> Thank you for your continued engagement and for raising these additional queries, which help us further clarify and strengthen our work. Our responses for each of the comments are provided below:
>
> > Comment 1 - No validation that classifier weights align with true source prototypes, nor that test-time prototypes remain close to them.
>
> Thank you for the insightful comment. To address your concern regarding the validation of our prototype-based design, we have added a t-SNE visualization in the revised manuscript (Fig.5 in Appendix). Specifically, we randomly selected 500 test-time prototypes ($p^c_t$) generated during the continual test-time adaptation process over different time-steps from the CIFAR-10-C dataset and plotted their t-SNE representation (blue dots). For comparison, we also included the source prototypes estimated from the classifier weights (red dots) and those computed directly from the source domain data (black stars). Here, the estimated source prototypes (red dots) correspond to classifier weights from the WideResNet-28 backbone as used during CIFAR-10 to CIFAR-10-C continual test-time adaptation, and these are the same as our initial prototype estimates ($p_0^c$ as used in Equation 3). The original source prototypes (black stars) correspond to the mean of features obtained by passing the CIFAR-10 source domain data through the same pretrained feature extractor (i.e. the entire model without the classifier layer) from the WideResNet-28 backbone, and prior works like RMT [2] or SANTA [3] use this method to obtain source prototypes.
>
> As shown in the figure (Fig.5 in Appendix), we obtain 10 clusters for 10 different classes of the CIFAR-10 dataset, and the estimated source prototypes, derived from the classifier weights are observed to be more or less well aligned with the true source prototypes obtained from the source data. Furthermore, the dynamic test-time prototypes (blue dots) too are observed to remain closely clustered around these estimated source prototypes. We believe this addition provides a strong empirical validation of our prototype-based design and the alignment of prototypes and helps in addressing your concern. We have also updated the same in sub-section A.6 of the manuscript.
>
> > Comment 2 -  Need for recalibration
>
> Yes we acknowledge that the performance improvement is marginal however not insignificant. Our experiments show that such simple recalibration technique when performed along with our dynamic $\alpha$ (CTA), the performance improvement is more than marginal _e.g._, in ImageNet-C the error rate is 59.9% when source prototype is fixed and CTA is not applied. The error rate drops to 58.1 when both are operational. This is a relative reduction of error of around 3%. Similarly the relative error reduction for CIFAR10-C and CIFAR100-C are 7.87% and 1% respectively.
>
> > Comment  3 - Highlight effectiveness of Symmetric Cross Entropy with Ablation
>
> Thank you for pointing this out. To validate the effectiveness of the symmetric cross-entropy (SCE) loss in our proposed DMSE framework, we conducted an ablation study by replacing the SCE loss with the standard categorical cross-entropy (CE) loss. We performed this experiment across three benchmark datasets -- ImageNet-C-5k, CIFAR-10C, and CIFAR-100C. As shown in the table below, in all three cases, we observed a consistent drop in performance when using CE in place of SCE, thereby demonstrating the importance and effectiveness of SCE in our setting. These results have now been included in our  manuscript (Appendix A.7).
>
>
> |Loss Type  | ImageNet-C-5k | CIFAR100-C | CIFAR10-C |
> | -------- | -------- | -------- | -------- |
> | CE    | 62.9  | 33.4 | 23.6 |
> | SCE  | 58.1 | 30.4  | 16.4 |

---

> ### Author Response · Authors · 2025-08-09
> **Response to Remaining Concerns (Part 2/2)**
>
> > Comment  4 - Generalizability of DMSE
>
> As shown in Table 2 of the manuscript DMSE outperforms most approaches while not requiring access to source dataset. However, in some scenarios, recent approaches like RDumb [1] perform better than DMSE. We performed extensive comparison of DMSE with RDumb. As shown in Table 1 and Table 2 of the manuscript, RDumb performs well in one among the 4 datasets (ImageNet-C-50k dataset). To further analyze how performance varies with the amount of test-time data, we provide a comparison in Table 3 of the manuscript (also reproduced below for convenience), which shows how both methods behave as the number of test samples increases. Our findings suggest that while RDumb performs good in longer test streams, DMSE shows stronger generalization in more challenging, low-data regimes. This property is particularly valuable for real-world deployment, where encountering novel conditions with limited data is a common challenge. We elaborate on this point in Section 4.1 (Comparison with RDumb), and below we duplicate the comparison of results from Tab. 3 for reference.
>
>
> | #Samples | 2.5k | 5k   | 7.5k | 10k  | 15k  | 25k  | 50k  |
> |----------|------|------|------|------|------|------|------|
> | **DMSE** | 59.5 | 58.1 | 57.9 | 57.8 | 57.7 | 57.6 | 57.5 |
> | RDumb    | 62.7 | 60.2 | 58.7 | 57.0 | 55.9 | 54.3 | 53.2 |
>
> *References:*
> 1. *Ori Press, Steffen Schneider, Matthias Kuemmerer, and Matthias Bethge. RDumb: A simple approach that questions our progress in continual test-time adaptation. In Thirty-seventh Conference on Neural Information Processing Systems, 2023.*
> 2. *Döbler, Mario, Robert A. Marsden, and Bin Yang. "Robust mean teacher for continual and gradual test-time adaptation." Proceedings of the IEEE/CVF Conference on Computer Vision and Pattern Recognition. 2023.*
> 3. *Goirik Chakrabarty, Manogna Sreenivas, and Soma Biswas. SANTA: Source Anchoring Network and Target Alignment for Continual Test Time Adaptation. Transactions on Machine Learning Research, 2023. ISSN 2835-8856.*

---

### Review · Reviewer_Wdov · 2025-06-26

**Summary Of Contributions:**

This paper proposes a controlled teacher adaptation (CTA) methodology for source-free continual test-time adaptation (CTTA). CTA dynamically adjusts the momentum value, used for updating a weight-averaged mean teacher to produce pseudo-labels, based on the quality of the incoming data. It further estimates class prototypes from the source pretrained model to help align the target data as they come in. The proposed approach outperforms different state-of-the-art adaptation methods or achieves comparable performance with them, without requiring access to source data.

**Audience:**

Yes

**Broader Impact Concerns:**

There is no concerns on the ethical implications.

**Claims And Evidence:**

Yes

**Requested Changes:**

Please address the questions in Cons and revise the manuscript accordingly.

**Strengths And Weaknesses:**

**Pros**:

+ This paper is generally well-organized and written, thus easy to follow.

+ The motivation of this paper is clear. Firstly, the motivation of adaptively setting momentum for mean teacher is clear, as demonstrated in Figure 1. Secondly, the motivation for using the weights of the classifier in the pre-trained model is well-presented and reasonable.

+ This paper proposes a dynamic momentum update based on the average prediction entropy, enabling the teacher to adapt to distribution shifts, leading to better CTTA performance.

+ This paper leverages the classifier’s weights to estimate source prototypes without requiring access to the source domain data during adaptation.

+ Experiments and ablations on multiple benchmark datasets show the benefits of DMSE over SOTA.


**Cons**:

1. Ablation on the resetting. Though Fig. 3(d) can somewhat clarify the necessity of the resetting technique, it would be better to provide the error rate of the setting without resetting.

2. Why does the dist function in Eq. (3) have a scale factor of 0.5? Please clarify the necessity of the class prototype first. Also, why is it necessary to update the class prototype as per Eq. (3) rather than directly updating the weights of the classifier via gradient descent? Is it because the classifier is fixed?

3. I believe that the weight of $L_{CL}$ in Eq. (6) should be properly tuned. In this case, please evaluate the sensitivity of its loss weight and the temperature $\tau$.

4. In Tab. 5, the Re-calibrating Class-wise Prototypes scheme seems less effective, often causing only a 0.3% gap in mean error. It would be better to verify its effectiveness with stronger experimental results.

5. In Tab. 4, the proposed method cannot beat RDumb or RMT. Please analy\e the possible reasons.

6. Does Tab. 9 imply that the proposed dynamic updating scheme for the prototypes is not ideal, as it can be worse than the static scheme? Is the Re-calibrating Class-wise Prototypes in Tab. 5 the same as the Dynamic $h_0$ in Tab. 9?

---

> ### Author Response · Authors · 2025-07-13
> **Response to Reviewer Wdov (Part 1/2)**
>
> We sincerely thank the reviewer for their positive and encouraging feedback on our motivation, method design, and empirical validation. Below are our responses to the specific concerns.
> >Ablation on the resetting. Though Fig. 3(d) can somewhat clarify the necessity of the resetting technique, it would be better to provide the error rate of the setting without resetting.
>
> Thanks for the suggestion. The intermediate resetting proves essential for model plasticity as indicated in Fig. 3(d). Following the reviewer’s suggestion, we carried out an experiment without the intermediate teacher-model resetting, which gave a poorer average error rate of 59.9% on the ImageNet-C-5k dataset as compared to our complete DMSE result (with the resetting technique active) of 58.1%.
>
> >Why does the dist function in Eq. (3) have a scale factor of 0.5? Please clarify the necessity of the class prototype first. Also, why is it necessary to update the class prototype as per Eq. (3) rather than directly updating the weights of the classifier via gradient descent? Is it because the classifier is fixed?
>
> The scale factor of 0.5 in the distance function of Eq. (3) is used to normalize the cosine distance to the [0,1] range. Since the cosine similarity between two normalized vectors ranges from -1 to 1, the transformation $0.5\times(1−cos⁡(⋅))$ ensures the resulting distance threshold is kept within the 0-to-1 range.
> The need for class prototypes arises because, in the absence of source data, they serve as stand-ins to represent class centers and facilitate feature alignment during adaptation. These prototypes are used specifically for computing the contrastive loss and not for direct classification. Regarding classifier updates, the classifier weights are not fixed and they are updated via gradient descent itself. Eq.(3) denotes the computation of class-wise prototypes which are used while computing the contrastive loss.
>
> >I believe that the weight of $L\_{CL}$ in Eq. (6) should be properly tuned. In this case, please evaluate the sensitivity of its loss weight and the temperature
>
> Thank you for the suggestion. We adopted the temperature value of 0.1 and the contrastive loss weight of 0.5 based on the findings of the RMT paper [1], which demonstrated their effectiveness in similar continual test-time adaptation settings. To further validate this choice, we conducted a sensitivity analysis on the ImageNet-C-5k dataset by varying both the temperature and the loss weight. Our experiments confirmed that the chosen values yield the best performance across corruption types. This analysis and the corresponding results have been added in the appendix of the revised version of the paper for completeness.
>
>
> |  | weight=0 | weight=0.25 | weight=0.75 | weight=1.0 |
> | -------- | -------- | -------- | -------- | -------- |
> | temperature=0.1     | 60.97  | 59.17  | 58.10   | 58.58  | 58.27  |
> | temperature=0.5  | 59.34  | 59.14  | 59.15  | 59.11  | 59.14  |
> | temperature=1.0  | 59.59  | 59.63  | 59.75  | 59.28  | 58.96  |
>
> >In Tab. 5, the Re-calibrating Class-wise Prototypes scheme seems less effective, often causing only a 0.3% gap in mean error. It would be better to verify its effectiveness with stronger experimental results.
>
> While the introduction of re-calibrated class prototypes may have brought about small changes, the improvements have been consistent across datasets and settings as observed from Tab. 5 and across the ablation experiments
>
> >In Tab. 4, the proposed method cannot beat RDumb or RMT. Please analyse the possible reasons.
>
> The experiments in Tab. 4 denote the average results over different sequences of changing corruptions. Since the 10 sequences used are randomly sampled orderings of the corruptions, the results too have some variability and may alter the relative performance across methods. DMSE achieves the best results on CIFAR100-C, while ranking the second best in other cases. Hence, DMSE performs consistently well across all datasets while also inducing less variability across different sequences as denoted in Fig. 3. a.
>
> *References*
> 1. *Döbler, Mario, Robert A. Marsden, and Bin Yang. "Robust mean teacher for continual and gradual test-time adaptation." Proceedings of the IEEE/CVF Conference on Computer Vision and Pattern Recognition. 2023.*

---

> ### Author Response · Authors · 2025-07-13
> **Response to Reviewer Wdov (Part 2/2)**
>
> >Does Tab. 9 imply that the proposed dynamic updating scheme for the prototypes is not ideal, as it can be worse than the static scheme? Is the Re-calibrating Class-wise Prototypes in Tab. 5 the same as the Dynamic in Tab. 9
>
> The experiments in Tab. 9 are for experiments carried out on prototype ‘centers’, and the 2 rows static $h\_0$ and dynamic $h\_0$ correspond to static prototype centers and dynamic prototype centers respectively . The prototype centers (denoted as $p^c$ in Eqn. 3) are used as anchors only during pseudo-label assignment i.e. the cosine distances of the test batch feature vectors are computed from these prototype centers. The re-calibration of class-wise prototypes in Tab. 5 denotes the updation/computation of the actual prototypes ($p^{c}_{t'}$ in Eq. 3) that are used while calculating the contrastive loss, while Tab. 9 denote the updation of the prototype centers which are used to assign pseudo labels to the test batch feature vectors. We have incorporated the updated explanation for the same in the corresponding 'Different ways of updating the prototypes' section under Ablation and additional analysis (Section 4.2).

---

> > ### Comment · Reviewer_Wdov · 2025-07-29
> > **Further discussion**
> >
> > Thanks for the response which has addressed most of my concerns. There are some minor comments for further improvement:
> >
> > 1. Please update the manuscript according to the response. The revision manuscript has only a few changes, which obviously cannot address all the concerns raised by the reviewers. For instance, the answer to Q1 Ablation on resetting says, “we carried out an experiment without the intermediate teacher-model resetting”. However, the results do not seem to be included in the revision or highlighted in the revision. Another example is that the possible reasons for the proposed method failing to beat RDumb or RMT have not been included in the revision.
> >
> > 2. Eq. (3) is hard to understand even after reading the response. For all $t’ > t$, $p^c_{t’}$ depends on $g_t$ and $p_c$. This means that when $t’=t+i, i=1,2,...,n$, $p^c_{t’}$ is decided by the $g_t$ at the time step $t$. If $i$ is a very large number, is it reasonable to use an old-fashioned $g_t$ to update the prototype at the current time step $t’=t+i$? Additionally, $p_c$ does not have a subscript $t$ or $t’$. Does it mean that $p_c$ is fixed, unchanged when $t$ varies?
> >
> > 3. The dynamic and static settings in Table 9 are still not clear. Does it mean that the $p_c$ in Eq. (3) adopts a static updating technique? However, the static and dynamic updating techniques are not well introduced.

---

> > > ### Author Response · Authors · 2025-08-02
> > > **Clarifications and Additional Discussion**
> > >
> > > Thank you for your detailed observations and constructive feedback. Our responses for each of the comments are provided below:
> > > > Comment 1 - Please update the manuscript according to the response.
> > >
> > > Thank you for your suggestion. We have updated the manuscript accordingly to reflect the changes mentioned in our response. In the updated manuscript we have incorporated the finding on the experiment without the intermediate teacher-model resetting in sub-section A.3 on page 18 and the explanation for possible reasons why our proposed method may not outperform RDumb or RMT in certain cases, as seen in Table 4, is added in sub-section 4.2 (page 10). In addition to these updates, we have also addressed comments raised by other reviewers, and the corresponding revisions have been incorporated throughout the manuscript. All modifications are highlighted in blue for clarity.
> > >
> > >
> > > > Comment 2 - Clarification on Prototype Updation
> > >
> > > Thanks for the detailed comment. We acknowledge and understand that the equation and the corresponding ablation experiment (table 9) had some gaps in terms of the confusing notations. We have, accordingly, updated the text (marked in blue, around equation 3 and around table 9). We request the reviewer to have a look at them. We are also explaining here briefly the edits and the process we follow for better understanding.
> > >
> > > Let us see if we are using an old-fashioned $g_t$ or not. We are not using old-fashioned $g_t$. This is because, we update the source prototypes $p^c_t$ at every timestep and this is used to compute the contrastive loss $\mathcal{L}_{CL}$ (defined in eqn. (6)) which, in turn, updates $g_t$ via backpropagation. Please note that we have changed the subscript from $t'$ to $t$ in the updated manuscript to be consistent with the notation $t$ as time for $g_t$ and $p^c_t$.
> > >
> > > Now let us come to the next part of the question which asks if $t' > t$ means $t' = t+i, i=1, 2, \cdots n$? Let us give the answer first without going into the notational changes. No, the gap between $t'$ and $t$ is not more than $1$. To make this more explicit, we now used $t-1$ as the subscript of $g$ in eqn. (3) when we are getting $p^c_t$ (the L.H.S.). Specifically, we use two time dependent entities to compute $p^c_t$ - i) $g_{t-1}$ and ii) $p^c_{t'}$ where $t' < t$. However, we experimented with two particular values of $t'$. They are $t'=t-1$ and $t'=0$ and found that $t'=0$ (*i.e.*, the initial estimated source prototypes from the classifier) works best. This is possibly due to the drift in later prototypes estimated from noisy pseudo-labels. We have expressed this finding in Table 9. In Table 9 also, we have made similar notational modifications to reflect this understanding.
> > >
> > > A **TLDR** to summarize can be - the updated prototypes $p^c_t$ at time instant $t$ are dependent on the feature extractor $g_{t-1}$ at time $t-1$ and on old source prototypes where 'old' can mean source prototypes at time $t-1$ (*i.e.*, at $t'=t-1$) or source prototypes estimated initially i.e., at $t'=0$.
> > > Hope this helps remove the confusion.
> > >
> > > > Comment 3 - Clarification on Ablation results reported in Table 9.
> > >
> > > We understand that the use of the term 'static' and 'dynamic' in this context can be confusing. What we mean by this is how we update $p^c_t$. Note that we always update $p^c_t$. So, it is not that $p^c_t$ is static. Rather, we meant that we used $p^c_0$ (source prototypes estimated initially from the classifier) to get the updated $p^c_t$ *i.e.*, $t'=0$ in eqn. (3). And the corresponding 'dynamic' update was another variation where we tried to see if we use the latest source prototype $p^c_{t-1}$ to update $p^c_t$ *i.e.*, $t'=t-1$ in eqn. (3). To clear the confusion we have got rid of the terms 'static' and 'dynamic' in this context and used $t'$ to denote which type of update is used. This is reflected in Table 9 of the revised manuscript (in blue color).

---

> > > > ### Comment · Reviewer_Wdov · 2025-08-02
> > > > **My concerns have been addressed**
> > > >
> > > > I appreciate the prompt response provided by the authors. The response has addressed my concerns on Eq. (3) and Table 9. The revision also includes new content to address most of the concerns of all the reviewers. I believe this paper is now ready for acceptance.

---

> > > > > ### Author Response · Authors · 2025-08-02
> > > > > **Response to Reviewer Wdov02 feedback**
> > > > >
> > > > > Thank you for the feedback and we are glad that our response addressed all the concerns.

---

### Review · Reviewer_7QZD · 2025-06-29

**Summary Of Contributions:**

The paper tackles the problem of Source-Free Continual Test-Time Adaptation (CTTA). After a model is deployed, the input domain can drift continuously over time, yet most prior methods either assume a single, static target domain or require access to the original (source) data. The authors propose a teacher–student framework called DMSE (Dynamic Momentum & Source Estimation). The key idea is to treat the pre-trained classifier weights as class prototypes, providing reference points without ever accessing source data. In parallel, the method adjusts the teacher’s EMA momentum on the fly, using the prediction entropy of each test mini-batch to curb pseudo-label errors and prevent teacher drift. Experiments on streaming benchmarks such as ImageNet-C, CIFAR-C, and DomainNet-126 show that DMSE generally surpasses existing state-of-the-art methods (e.g., RMT, SANTA) without any source data, and adapts especially quickly when only a handful of test samples are available.

**Audience:**

Yes

**Claims And Evidence:**

Yes

**Requested Changes:**

- Include experiments that track error accumulation over long streams under both fixed and dynamic momentum to substantiate the claimed protection against teacher collapse.
- Discuss how the approach might generalize to tasks where classifier weights do not naturally align with class prototypes, such as object detection or semantic segmentation.
- Clearly describe which components of DMSE are genuinely novel, how they differ from prior prototype-based or adaptive-momentum methods, and provide ablation studies demonstrating that each new element is essential to the observed performance gains.

**Strengths And Weaknesses:**

Strengths

Rapid initial adaptation:
- By reusing the pre-trained classifier weights as initial prototypes, DMSE achieves low error rates with very few samples, making it practical for real-world deployments where data is scarce.


Weaknesses

Architectural scope:
- The prototype trick hinges on a linear classification head; tasks that do not employ such heads (e.g., object detection, segmentation) may not benefit without further modification.

Limited evidence on teacher collapse:
- Although dynamic momentum is claimed to mitigate teacher collapse, the paper lacks concrete experiments contrasting fixed-α momentum versus the proposed dynamic schedule over long streams.

Insufficient novelty discussion:
- Initializing class prototypes with pre-trained classifier weights has already been explored (e.g., Lee, Hojoon, et al. "Prototypical class-wise test-time adaptation." Pattern Recognition Letters 187 (2025): 49-55.), and operating in a source-free regime is the default premise of CTTA. To justify novelty, the paper must specify precisely what is new in its prototype usage or dynamic-momentum scheme, and provide ablations or comparisons that show why these refinements outperform prior work.

---

> ### Author Response · Authors · 2025-07-13
> **Response to Reviewer 7QZD**
>
> We thank the reviewer for acknowledging the strength of our method in enabling rapid adaptation with limited data. We appreciate your positive feedback on the practicality of reusing pre-trained classifier weights for effective low-shot adaptation. Below are our responses to the specific concerns.
>
> >(Point 3 in Strengths and Weaknesses and Point 1 in Requested Changes) Include experiments that track error accumulation over long streams under both fixed and dynamic momentum to substantiate the claimed protection against teacher collapse.
>
> Thank you for the suggestion. To address the concern regarding error accumulation and the risk of teacher collapse over extended adaptation streams, we conducted additional experiments on the ImageNet-C-50k dataset, which contains 50,000 images per corruption domain. We evaluated our method using both fixed and dynamic momentum settings across these long sequences. The fixed momentum approach gives a 70.2% error rate while our approach with dynamic momentum gives an error rate of 57.5% on this ImageNet-C-50k sequence.
> This significant improvement demonstrates that dynamic momentum not only enhances adaptability over long streams but also provides effective protection against error accumulation and teacher collapse, thereby highligting the robustness of DMSE in continual adaptation scenarios
>
> > (Point 2 in Strengths and Weaknesses and Point 2 in Requested Changes) Discuss how the approach might generalize to tasks where classifier weights do not naturally align with class prototypes, such as object detection or semantic segmentation
>
> While our focus in this work is on classification, the proposed approach can be extended to tasks like semantic segmentation and object detection. Specifically, semantic segmentation can be viewed as dense per-pixel classification, where our prototype-based formulation remains applicable. When source data is available, initial prototypes can be computed from labeled data (e.g., as done in SANTA) and dynamically refined using our momentum-based updates. This has been shown in, prior works such as DIGA [2] which used instance-level prototypes for adapting segmentation models to distribution shifts at test time. We keep this as an option to explore in future.
>
> > (Point 4 in Strengths and Weaknesses and Point 3 in Requested Changes) Clearly describe which components of DMSE are genuinely novel, how they differ from prior prototype-based or adaptive-momentum methods, and provide ablation studies demonstrating that each new element is essential to the observed performance gains.
>
> Our work introduces two key innovations in the DMSE framework that set it apart from prior prototype-based and adaptive-momentum methods:
> 1. Dynamic Momentum for Teacher-Student Adaptation:
> Unlike previous approaches that use a fixed momentum parameter for updating the teacher model, DMSE employs an adaptive momentum mechanism. This allows the teacher to selectively incorporate knowledge from the student based on the confidence (entropy) of the student’s predictions, ensuring stable and effective adaptation throughout evolving target domains.
> 2. Dynamic Source Distribution Estimation:
> DMSE estimates the source distribution using dynamically updated class-wise prototypes, without requiring access to any original source domain data. While earlier works like [1] on source-free domain adaptation (SFDA) have explored surrogate source representations, they typically assume access to the entire target domain at once. In contrast, DMSE is designed for scenarios where target data arrives sequentially in batches, necessitating continual prototype recalibration as detailed in Section 3.2.
>
> Ablation studies, as summarized in Table 5, demonstrate that both the adaptive momentum mechanism and dynamic class-wise prototypes have significant contributions in the continual adaptation process. The combined approach yields an average error rate reduction of approximately 1.2% points across CTTA datasets, underscoring the necessity and effectiveness of each novel element in DMSE.
>
> *References:*
> 1. *Ning Ding, Yixing Xu, Yehui Tang, Chao Xu, Yunhe Wang, and Dacheng Tao. Source-Free Domain Adaptation via Distribution Estimation. IEEE Conference on Computer Vision and Pattern Recognition, pp. 7202–7212, 2022.*
> 2. *Wang, Wei, et al. "Dynamically instance-guided adaptation: A backward-free approach for test-time domain adaptive semantic segmentation." Proceedings of the IEEE/CVF Conference on Computer Vision and Pattern Recognition. 2023.*

---

> > ### Author Response · Authors · 2025-08-02
> > **Follow-Up on Rebuttal Submission**
> >
> > Dear reviewer,
> >
> > We sincerely thank all your efforts and apologize for sending out this note regarding our paper. We have addressed all the thoughtful questions and suggestions raised in the review and have updated the paper/rebuttal. We will be glad to have further discussion to see if our response solves the concerns. Thank you for your service!
> >
> > Best wishes,
> >
> > Authors

---

### Decision · Action_Editor_Wnop · 2025-10-05

**Recommendation:** Reject

**Additional Comments:**

This work targets an important CTTA problem and the idea is promising, but the current evidence does not support the core claims. The reviewers agreed that the prototype assumption is insufficiently validated and the reported gains are inconsistent against recent baselines. A revision focusing on explicit prototype-centroid alignment tests, stronger long-stream evaluations, and clear ablations of the loss components would substantially strengthen the case. Resubmission is encouraged once these issues are addressed and the new experiments are integrated into the main text.

**Audience:**

Yes

**Audience Explanation:**

Source-free continual test-time adaptation is of broad interest, and the paper’s direction of entropy-aware teacher updates and prototype guidance addresses a timely problem. However, interest alone is not enough to overcome the present shortcomings in validation and clarity.

**Claims And Evidence:**

No

**Claims Explanation:**

The core prototype assumption is not rigorously validated: there is no quantitative alignment test between classifier weights, true source centroids, and evolving test-time prototypes, nor tracking of per-class drift over time. Prototype recalibration yields marginal gains, leaving its necessity unclear. The loss design remains under-justified: SCE is borrowed from prior work without thorough ablations (interaction with the projection head, temperature/weight sensitivity), and Eq. (3) suffers from ambiguous indexing and the “prototype” vs. “prototype center” distinction. Evidence for long-stream robustness is incomplete: fixed vs. dynamic momentum comparisons and the effect of resets are not clearly established. Competitiveness is inconsistent against strong recent baselines (e.g., RDumb) and coverage of RoTTA/PETAL and ViT backbones is limited; ImageNet-C-5k and 50k are not cleanly separated.

**Resubmission Of Major Revision:**

The authors may consider submitting a major revision at a later time.